# Decoupled DMD: CFG Augmentation as the Spear, Distribution Matching as the Shield

**Dongyang Liu**[1,2]  **Peng Gao**[1,4,✉]  **David Liu**[1,2]  **Ruoyi Du**[1]  **Zhen Li**[1]  **Qilong Wu**[1]
**Xin Jin**[1]  **Sihan Cao**[1]  **Shifeng Zhang**[1]  **Steven HOI**[1]  **Hongsheng Li**[2,3,✉]

[1]Tongyi Lab, Alibaba Group    [2]MMLab, CUHK    [3]CPII under InnoHK
[4]Shenzhen Institutes of Advanced Technology, Chinese Academy of Sciences
✉jingpeng.gp@alibaba-inc.com    ✉hsli@ee.cuhk.edu.hk
 https://github.com/Tongyi-MAI/Z-Image

## Abstract

Diffusion model distillation has emerged as a powerful technique for creating efficient few-step and single-step generators. Among these, Distribution Matching Distillation (DMD) and its variants stand out for their impressive performance, which is widely attributed to their core mechanism of matching the student's output distribution to that of a pre-trained teacher model. In this work, we challenge this conventional understanding. Through a rigorous decomposition of the DMD training objective, we reveal that in complex tasks like text-to-image generation, where CFG is typically required for desirable few-step performance, the primary driver of few-step distillation is not the distribution matching term, but a previously overlooked component we identify as *CFG Augmentation* (**CA**). We demonstrate that this term acts as the core "engine" of distillation, while the **D**istribution **M**atching (**DM**) term functions more as a "regularizer" that ensures training stability and mitigates artifacts. We further validate this decoupling by demonstrating that while the DM term is a highly effective regularizer, it is not unique; simpler non-parametric constraints or GAN-based objectives can serve the same stabilizing function, albeit with different trade-offs. This decoupling of labor between CA and DM also allows a more principled analysis of the properties of both terms, leading to a more systematic and in-depth understanding. This new understanding enables us to propose principled modifications to the distillation process, such as decoupling the noise schedules for the engine and the regularizer, leading to further performance gains.

## 1 Introduction

Diffusion models (Sohl-Dickstein et al., 2015; Ho et al., 2020; Song et al., 2020) have rapidly risen to prominence in generative modeling, achieving state-of-the-art performance and producing images of unprecedented quality and diversity. This success has sparked widespread interest across both academia and industry. However, the power of these models comes at a significant cost: their iterative sampling process, often requiring dozens to hundreds of neural network evaluations, is computationally expensive and slow, hindering their use in real-time applications.

To address this limitation, a flurry of research has focused on converting the original diffusion model into a few-step generator. Typical technical routes include *direct distillation* (Liu et al., 2022), where the student is expected to replicate the trajectory of the teacher; *consistency distillation* (Song et al., 2023), which enforces self-consistency along the sampling trajectory; and *adversarial distillation* (Sauer et al., 2024b), which leverages an adversarial objective to match the student's output distribution to target, showing impressive results in high-resolution synthesis.

Among these diverse approaches, score-based distillation, notably represented by Diff-Instruct (Luo et al., 2023b), Distribution Matching Distillation (DMD) (Yin et al., 2024b), and other variants (Yin et al., 2024a; Chadebec et al., 2025), has been recognized as exceptionally promising. Its advantages are twofold: not only does it achieve state-of-the-art performance, but it is also underpinned by an elegant theoretical framework. Specifically, the method is framed as minimizing an Integral Kullback-Leibler (IKL) divergence (Luo et al., 2023b) between the student's output distribution

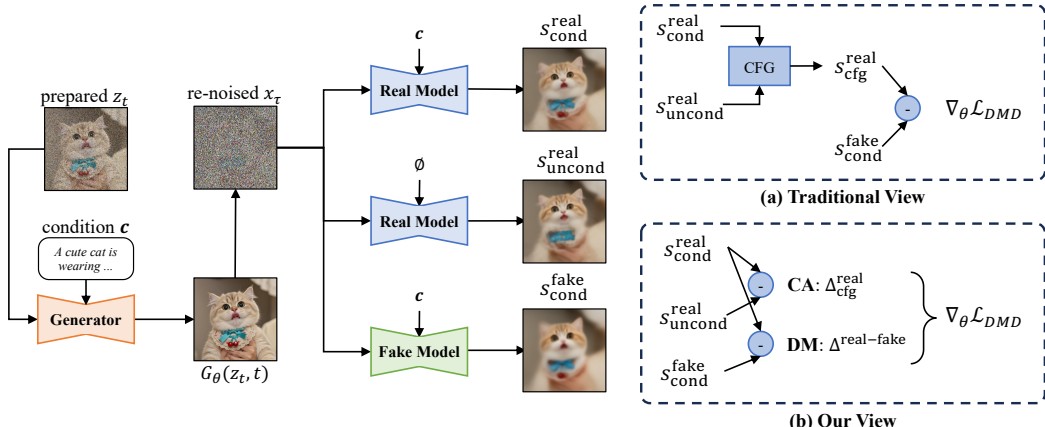

Figure 1: Two perspectives on the DMD algorithm. (a) The conventional view, which treats the use of CFG as a heuristic relaxation of the theoretical framework, with the algorithm's success solely attributed to this (relaxed) distribution matching mechanism. (b) Our proposed decoupled view, where the objective is a combination of two distinct mechanisms: a CFG Augmentation (CA) engine that drives the few-step conversion, and a Distribution Matching (DM) regularizer—which strictly adheres to the theoretical derivation (Eq. 1)—that ensures training stability.

($p_{\text{fake}}$) and the teacher's target distribution ($p_{\text{real}}$):

$$\mathcal{L}_{\text{IKL}}(p_{\text{real}}, p_{\text{fake}}) = \int_0^1 \mathbb{KL}(p_{\text{real},\tau} \,||\, p_{\text{fake},\tau}) \, d\tau. \tag{1}$$

In practice, the gradient of this objective is estimated using a pair of "real" (teacher) and "fake" (student-tracking) models, making the training process feasible.

**However, a dark cloud has loomed over the interpretation of this elegant method: the use of Classifier-Free Guidance (CFG) (Ho & Salimans, 2022) on the real model**. According to the theoretical derivation, the ideal estimator for the target real score should be the prediction from the real model itself, with no involvement of CFG. However, empirical evidence overwhelmingly shows that on complex tasks like text-to-image, DMD-like methods yield good results only with a high CFG scale. Even if we were to boldly assume—*an assumption we find to be insufficiently grounded*—that CFG somehow produces a higher-quality substitute for the original score, the *asymmetric* application of CFG, in which only the real model but not the fake model is equipped with CFG, still creates a stark inconsistency between theory and practice. Overall, the usage of CFG breaks the integrity of the original, rigorous theoretical derivation of matching two distributions.

This strongly suggests that **the current understanding of DMD's success is likely incomplete or inaccurate**. In this paper, we aim to redefine the understanding of how DMD and similar algorithms work. Through a rigorous decomposition of the practical DMD training objective that utilizes real score CFG, we reveal that its effectiveness is not driven by a single mechanism, but by a clear division of labor between two distinct components:

**1. CFG Augmentation (CA):** A previously overlooked term that directly applies the CFG signal to the student's output. We demonstrate that this component acts as the core **engine** of distillation, responsible for converting a multi-step model into a high-quality few-step generator.

**2. Distribution Matching (DM):** A mechanism that perfectly aligns with the theoretical derivation (Eq. 1). While existing works have proved its independent distillation capability in simple tasks like low-resolution CIFAR (Luo et al., 2023b), we show that for complex tasks, its primary function converts more to a powerful **regularizer** that ensures training stability and mitigates artifacts.

This decoupled framework challenges the prevailing narrative and provides a more accurate explanation for the success of DMD-like methods. We substantiate our claims through a series of carefully designed experiments, including independent investigations of the effect of each component, and demonstrating that the DM regularizer, while highly effective, can be conceptually replaced by simpler statistical constraints or more complex GANs. This explicit decoupling also enables

a more principled and in-depth analysis of the properties and inner workings of each component. Finally, armed with this deeper understanding, we propose principled improvement by proposing decoupled renoising schedules for CA and DM, respectively, leading to further performance gains, and demonstrating the practical value of our new perspective.

## 2 RELATED WORK

**Few-Step Diffusion Distillation** aims to reduce the inference cost of diffusion models. *Trajectory-matching* approaches train a student model to replicate the teacher's sampling path in fewer steps (Liu et al., 2023; Zhu et al., 2024; Kim et al., 2024; Frans et al., 2024; Salimans & Ho, 2022; Meng et al., 2023), with consistency distillation as a renowned branch (Song et al., 2023; Kim et al., 2023; Lu & Song, 2024; Wang et al., 2024; Ren et al., 2024). Another prominent direction is *GAN-based distillation* (Sauer et al., 2024b;a; Lin et al., 2024), which leverages an adversarial objective to match the student's output distribution with the teacher's or with real data.

**Score-based Distillation** was initially proposed for 3D generation (Poole et al., 2022; Wang et al., 2023). Diff-Instruct (Luo et al., 2023b) pioneered its application in few-step diffusion distillation, and DMD (Yin et al., 2024b) was among the first to successfully apply this principle to large-scale text-to-image models. Following works have explored different distribution metrics (Zhou et al., 2024b;a) or combining this principle with other distillation paradigms (Yin et al., 2024a; Chadebec et al., 2025; Luo et al., 2024). Notably, the adoption of CFG in real score is a common practice among these works, but this choice is rarely officially discussed. An exception is (Luo, 2024), which models the CFG term as an extra reward function after distillation. We are the first to decouple the role of this CFG term during distillation and to reveal its dominance in multi-to-few-step conversion.

## 3 REVISITING AND DECOMPOSING DMD

The goal of Distribution Matching Distillation (DMD) is to train a student generator, denoted as $G_\theta$, to emulate the output distribution of a pre-trained, frozen teacher diffusion model in a few-step or even single-step inference process. The training is guided by minimizing a loss function, Eq. 1, whose gradient with respect to the generator's parameters $\theta$ can be estimated by:

$$\nabla_\theta \mathcal{L}_{\text{DMD-theory}} = \mathbb{E}_{z_t,\tau,\mathbf{x}_\tau} \left[ -\left( s_{\text{cond}}^{\text{real}}(\mathbf{x}_\tau) - s_{\text{cond}}^{\text{fake}}(\mathbf{x}_\tau) \right) \frac{\partial G_\theta(z_t)}{\partial \theta} \right]. \tag{2}$$

**In this paper, we follow the flow matching notations (Lipman et al., 2022) and define $t = 0$ with pure noise and $t = 1$ with clean data.** $z_t$ denotes the prepared generator input at noise level $t$. For single-step generation, $t$ is 0 and $z_t$ is random noise; for few-step generation, $z_t$ can be obtained by going through the previous steps, a technique called "backward simulation" (Yin et al., 2024a). The generator $G_\theta$ takes $z_t$ and makes the image prediction $G_\theta(z_t)$, which is then *renoised* to $x_\tau$ with a sampled noise level $\tau$. After renoising, $x_\tau$ would be fed to two diffusion models for score estimates: $s_{\text{cond}}^{\text{real}}$, the "real score" estimated by the original multi-step teacher model; and $s_{\text{cond}}^{\text{fake}}$, the "fake" score estimate from an auxiliary "fake" model that is trained concurrently on the generator's outputs. The subscript "cond" indicates the score is conditioned on a text input. Pseudo-code provided in Sec. C

However, while Eq. 2 proved to be effective on simple generation tasks, its performance on complex tasks like text-to-image is usually unsatisfactory, and a subtle modification is required:

$$\nabla_\theta \mathcal{L}_{\text{DMD}} = \mathbb{E}_{z_t,\tau,\mathbf{x}_\tau} \left[ -\left( s_{\text{cfg}}^{\text{real}}(\mathbf{x}_\tau) - s_{\text{cond}}^{\text{fake}}(\mathbf{x}_\tau) \right) \frac{\partial G_\theta(z_t)}{\partial \theta} \right]. \tag{3}$$

The only difference between Eq. 2 and Eq. 3 is that the real score $s_{\text{cond}}^{\text{real}}$ is replaced with $s_{\text{cfg}}^{\text{real}}$, where

$$s_{\text{cfg}}^{\text{real}}(\mathbf{x}_\tau) = s_{\text{uncond}}^{\text{real}}(\mathbf{x}_\tau) + \alpha \left( s_{\text{cond}}^{\text{real}}(\mathbf{x}_\tau) - s_{\text{uncond}}^{\text{real}}(\mathbf{x}_\tau) \right). \tag{4}$$

$s_{\text{cond}}^{\text{real}}$ and $s_{\text{uncond}}^{\text{real}}$ are the conditional and unconditional scores from the real model, respectively, and $\alpha$ is the CFG guidance scale (typically $\alpha > 1$). Despite the introduction of discrepancy between theory and practice, this modification empirically yields substantially better results. Interestingly, this substitution has been largely overlooked in prior literature, often dismissed as a mere implementation detail rather than a fundamental deviation from the original theory. **However, we will show**

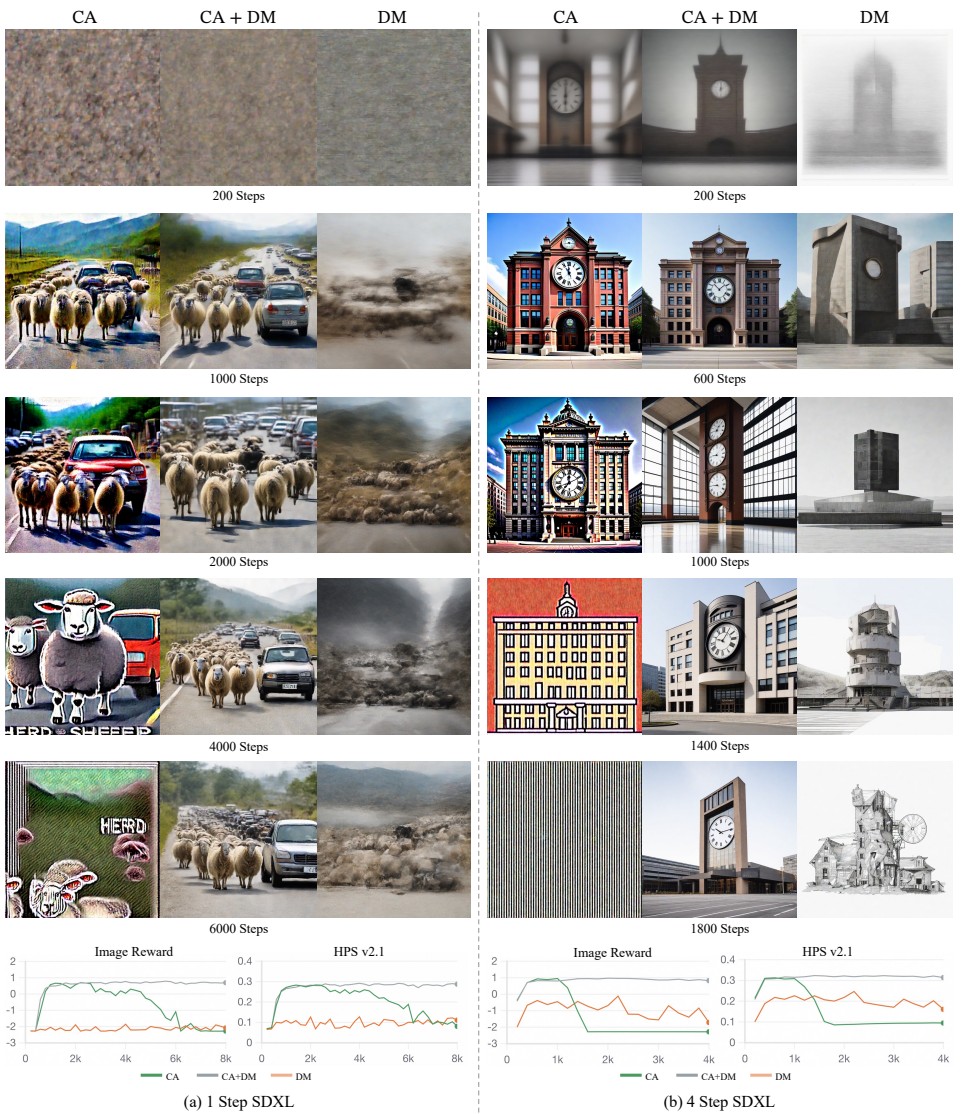

Figure 2: Ablation study on the roles of CFG Augmentation (CA) and Distribution Matching (DM). Numerical indicators are evaluated on 1k sampled prompts from COCO-10k (Lin et al., 2014).

**that the seemingly minor CFG detail actually signifies a fundamentally different mechanism independent to distribution matching.**

For clarity, in the rest of the paper, we use the acronym "DMD" to specially refer to in-practice-used and CFG-involved algorithm as defined in Eq. 3. In contrast, we use the term "Distribution Matching" or its abbreviation "DM" to refer exclusively to the theoretical principle of matching two distributions, which strictly adheres to the formulation in Eq. 1 and Eq. 2. We will show that the success of DMD is from the cooperation of two different mechanisms, with Distribution Matching playing a crucial, yet secondary, role as a regularizer rather than the primary distillation engine.

### 3.1 DECOMPOSING THE DMD GRADIENT

To scrutinize the underlying mechanisms of the DMD algorithm, we begin by substituting the definition of Classifier-Free Guidance (Eq. 4) into the DMD gradient formula (Eq. 3):

$$\nabla_\theta \mathcal{L}_{\text{DMD}} = \mathbb{E}\left[ -\left[ s_{\text{uncond}}^{\text{real}}(\mathbf{x}_\tau) + \alpha \left( s_{\text{cond}}^{\text{real}}(\mathbf{x}_\tau) - s_{\text{uncond}}^{\text{real}}(\mathbf{x}_\tau) \right) - s_{\text{cond}}^{\text{fake}}(\mathbf{x}_\tau) \right] \frac{\partial G_\theta(z_t)}{\partial \theta} \right]. \tag{5}$$

With simple rearrangement, we can decompose Eq. 5 into two distinct components:

$$\nabla_\theta \mathcal{L}_{\text{DMD}} = \mathbb{E}\left[ -\left( \underbrace{\left( s_{\text{cond}}^{\text{real}}(\mathbf{x}_\tau) - s_{\text{cond}}^{\text{fake}}(\mathbf{x}_\tau) \right)}_{\Delta^{\text{real-fake}} \text{ (Distribution Matching)}} + (\alpha - 1) \underbrace{\left( s_{\text{cond}}^{\text{real}}(\mathbf{x}_\tau) - s_{\text{uncond}}^{\text{real}}(\mathbf{x}_\tau) \right)}_{\Delta_{\text{cfg}}^{\text{real}} \text{ (CFG Augmentation)}} \right) \frac{\partial G_\theta(z_t)}{\partial \theta} \right]. \tag{6}$$

This decomposition reframes the DMD objective as a sum of two terms:

**1. Distribution Matching (DM, $\Delta^{\text{real-fake}}$):** The first term, $s_{\text{cond}}^{\text{real}} - s_{\text{cond}}^{\text{fake}}$, strictly aligns with theoretical deduction of matching two distributions (Eq. 1 and 2).

**2. CFG Augmentation (CA, $\Delta_{\text{cfg}}^{\text{real}}$):** The second term, $(\alpha - 1)(s_{\text{cond}}^{\text{real}} - s_{\text{uncond}}^{\text{real}})$, directly applies a scaled CFG signal as a gradient to the student's output. This component was typically overlooked.

This separation motivates an ablation study to isolate the true contribution of each component. We investigate three training configurations: (1) the full DMD objective ($\Delta^{\text{real-fake}} + \Delta_{\text{cfg}}^{\text{real}}$), (2) CFG Augmentation only ($\Delta_{\text{cfg}}^{\text{real}}$), and (3) Distribution Matching only ($\Delta^{\text{real-fake}}$).

### 3.1.1 ABLATION STUDY: ENGINE VS. REGULARIZER

As illustrated in Fig. 2, our experiments reveal a clear division of labor between the two components. Training with CA alone is remarkably effective at converting the multi-step model into a few-step generator. Besides, the generated results also demonstrate high similarity in content to the full DMD objective, indicating the dominant role of the CA term in DMD loss. In contrast, even though it is improper to conclude that the DM term is totally incapable of doing the multi-step to few-step conversion (since in the 4-step experiment it indeed makes relatively reasonable images), a significant performance gap exists towards the CA setting, as indicated by both image visualizations and numerical indicators (Image Reward (Xu et al., 2023) and HPS v2.1 (Wu et al., 2023)).

However, we also observe that training with CA alone is unsustainable. While initially effective, the generated images progressively suffer from artifacts such as over-saturation and high-frequency noise, eventually leading to training collapse. The introduction of the Distribution Matching term eliminates these issues, enabling stable training over extended periods and yielding higher-quality final results. These empirical findings lead to two fundamental conclusions:

**1. CFG Augmentation is the engine for few-step conversion.** The ability of the distilled generator to produce high-quality samples in a few steps is almost entirely attributable to the $\Delta^{\text{cfg}}$ term.

**2. Distribution Matching is a regularizer for training stability.** The $\Delta^{\text{real-fake}}$ term, while not the primary driver of distillation, plays a crucial role as a regularizer that prevents the training process from diverging and ensures the quality of the final output.

This insight fundamentally challenges the prevailing understanding of DMD-like methods: the conversion to a few-step generator is not primarily an act of matching distributions but rather **a direct consequence of "baking" the CFG pattern into the student generator's predictions** (we elaborate on this point in Sec. B), which is irrelevant to the fake model.

### 3.2 DISTRIBUTION MATCHING: A GOOD, BUT NOT THE ONLY, REGULARIZER

To further validate the aforementioned division between engine and regularizer, we investigate whether alternative regularization schemes can effectively stabilize the CFG Augmentation (CA) engine. This section demonstrates that even a simple non-parametric statistical constraint can prevent training collapse, thereby substantiating the role of Distribution Matching (DM) as a regularizer. Concurrently, we highlight that DM is an exceptionally well-suited regularizer for this task, exhibiting clear advantages over both simpler non-parametric methods and more complex GAN-based approaches.

**Non-Parametric Mean-Variance Regularization.** As shown in Fig. 3, training with the CA engine leads to a monotonic increase in the variance of generated images, finally reaching unreasonably large values. This inspires us to design the simplest regularization term of constraining the mean and variance of the generator's output. Specifically, we apply a Kullback-Leibler (KL) divergence loss that aligns the per-image mean $\mu$ and variance $\sigma^2$ of the student's output $G_\theta(z_t)$ with target statistics $(\mu_{\text{target}}, \sigma_{\text{target}}^2)$. For SDXL experiments, we use $\mu_{\text{target}} = 0.075$ and $\sigma_{\text{target}}^2 = 0.81$, which are

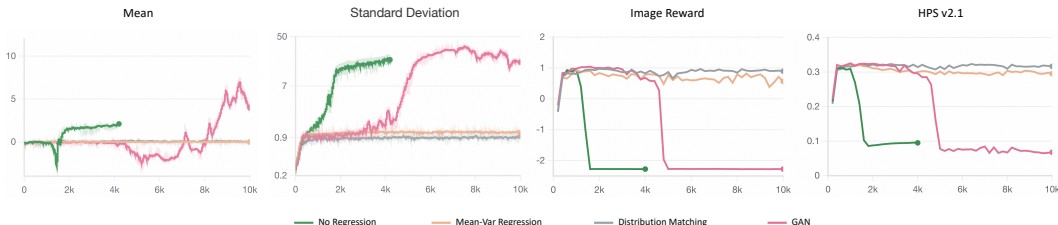

Figure 3: CFG Augmentation with different regularizers. Image Reward and HPS v2.1 evaluated on 1k sampled prompts from COCO-10k. Setting: 4-step SDXL. See Fig. 8 for visualized samples.

the averaged statistics of sampled real data. For a batch of $B$ generated images, the loss is defined as:

$$\mathcal{L}_{\text{KL}} = \frac{1}{B} \sum_{i=1}^{B} \frac{1}{2} \left( \frac{\sigma_i^2 + (\mu_i - \mu_{\text{target}})^2}{\sigma_{\text{target}}^2} - 1 - \log \frac{\sigma_i^2}{\sigma_{\text{target}}^2} \right), \tag{7}$$

where $(\mu_i, \sigma_i^2)$ are the mean and variance of the $i$-th generated image.

As shown in Fig. 3 and Fig. 8, this simplest non-parametric regularization proves remarkably effective at stabilizing the training process, allowing the CA engine to operate durably, keeping the quality indicators at a relatively high level. This result strongly reinforces the hypothesis that the primary role of the DM term is a regularizer. However, the final image quality, while stable, falls noticeably short of that achieved with DM. This suggests that the artifacts induced by the CA engine are more complex than what can be captured by mean&variance alone.

**GAN-based Regularization** A more powerful candidate for regularization is a GAN discriminator, a technique already employed in several diffusion distillation works (Yin et al., 2024a; Lin et al., 2025). Following existing methods (Sauer et al., 2024b), we use a discriminator initialized from the weights of the pre-trained teacher model. Experiments (Fig. 3) confirm that a GAN can indeed function as a regularizer, effectively controlling image variance and eliminating certain artifacts. However, GAN requires image data during the distillation process. Besides, the approach introduces significant challenges in training stability, as the training still collapsed after 4k iterations.

Our investigation suggests that while distribution matching is not the only possible regularizer, it represents a sweet spot—offering a more powerful corrective signal than simple statistical constraints, while being substantially more stable and less complex than GANs. The comparison suggests a trade-off between stability and potential performance. The increasing complexity from statistical constraints to GANs correlates with decreasing training robustness, which echoes the claim in Diff-Instruct (Luo et al., 2023b) that score-matching can be viewed as a more stable alternative to GANs, especially when the distributions have disjoint supports. Conversely, greater complexity may offer a higher performance ceiling. This aligns with practices in VAE training (Rombach et al., 2021; 2022) or advanced few-step distillation (Lin et al., 2025), where models are often first trained with a stable objective before being fine-tuned with a GAN loss to achieve peak performance.

## 4 MECHANISTIC ANALYSIS OF CA AND DM

The decomposition of the DMD objective into a CA engine and a DM regularizer enables an in-depth exploration of their respective inner workings. This section aims to answer two fundamental questions: first, how exactly does the CA engine drive the multi-step to few-step conversion? And second, by what mechanism does the DM regularizer ensure the stability of this process?

### 4.1 DISSECTING THE CA ENGINE: THE ROLE OF THE RE-NOISING SCHEDULE

To understand the working mechanism of the CA engine, we investigate a central question: **How does the choice of re-noising timestep $\tau$ influence the effect of CA**? Since $\tau$ determines the noise level at which the CFG signal is computed, it serves as a powerful probe into the engine's behavior.

To isolate this effect, we design an experiment where a single-step generator is trained using only the CA term ($\Delta_{\text{cfg}}^{\text{real}}$). We then systematically vary the sampling range of the re-noising timestep $\tau$, starting from the noisiest end of the spectrum and gradually expanding towards cleaner timesteps.

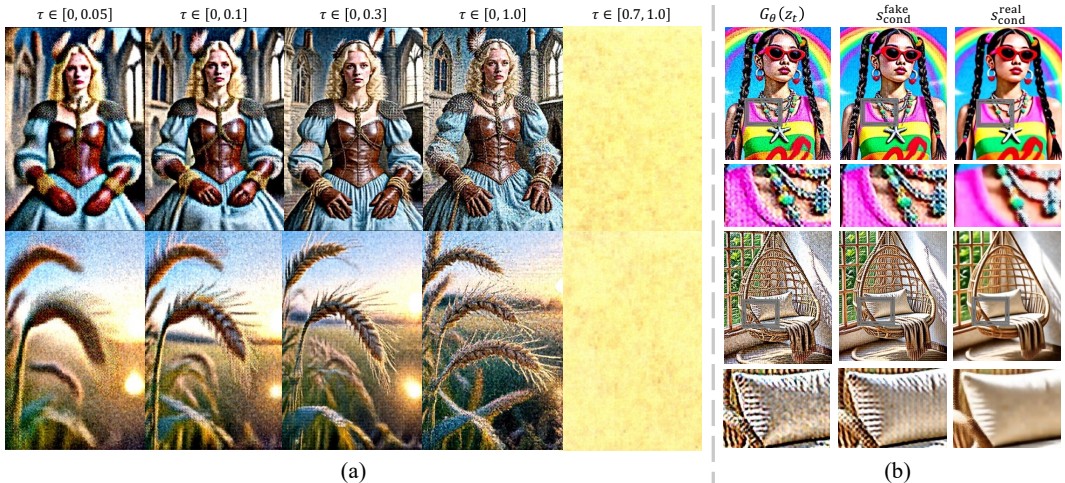

Figure 4: (a) Visualization on the effect of re-noising timestep $\tau$ in CFG Augmentation (CA). The generator is trained with CA alone. In our notation, $\tau = 0$ corresponds to pure noise and $\tau = 1$ to clean data. (b) Illustration of the DM corrective mechanism. The generator is trained with CA alone, while the fake model keeps training on the generator's output as in DMD.

Results in Fig. 4 (a) reveal a clear and consistent pattern. When $\tau$ is restricted to a noisy range (e.g., $[0.0, 0.05]$), the CA engine primarily enhances low-frequency information, such as broad color blocks and overall composition. As the range of $\tau$ expands to include cleaner timesteps, the generated images progressively gain richer, higher-frequency details, like sharp edges and fine textures. This leads to a crucial conclusion: **the CA engine, when applied at a specific noise level $\tau$, primarily enhances the image content corresponding to that level**. This conclusion is further supported by the fact that the training collapses when $\tau$ is restricted to clean timesteps only ($[0.7, 1.0]$): high-frequency details are meaningless if low-frequency general structure has not yet been determined.

This finding has a critical implication for multi-step generation. Consider a generator at step $t$ operating on an input $z_t$, which already contains resolved information for noise levels below $t$. Is it still necessary, or even detrimental, to apply CA with a re-noising range containing $\tau < t$? Our analysis suggests this would be redundant, potentially over-enhancing already established features and leading to artifacts. We therefore hypothesize that **an optimal CA schedule should act as a focused engine, concentrating its power on the remaining, unresolved aspects of the image by constraining its re-noising schedule to $\tau > t$.** Experimental validation is provided in Sec. 4.3.

## 4.2 Understanding the DM Regularizer: A Corrective Mechanism

Having established CA as the engine, we now turn to the DM regularizer and ask: **How does it counteract the artifacts introduced by CA and ensure training stability?**

To gain insight into this corrective mechanism, we design a specific diagnostic experiment. We continue to train the generator using only the CA engine, a setting we know is unstable and produces artifacts. However, we introduce an auxiliary "fake" model as a non-interfering **observer**. This fake model is trained concurrently on the generator's outputs—just as in standard DMD—but its score estimate is **not** used to update the generator. This setup allows us to witness *when artifacts occur, how a potential DM gradient ($s_{cond}^{real} - s_{cond}^{fake}$) would act to correct them.*

Fig. 4 (b) offers an informative observation. The image generated by the CA-only generator exhibits clear high-frequency checkerboard artifacts. When this artifact-laden image is re-noised and fed to the two score models, the artifact is conspicuously absent in the prediction from the frozen real model, yet it persists in the prediction from the observer fake model. This occurs because the fake model, by tracking the generator's output distribution, has learned to replicate its characteristic failure modes. Consequently, in the DM gradient, the artifacts present in the fake prediction ($s_{cond}^{fake}$) form a negative term. Applying this gradient to the generator's output would thus encourage a change that actively cancels out these artifacts. This provides a concrete illustration of DM's corrective mechanism.

Table 1: Ablation on different re-noising schedule configurations with Lumina-Image-2.0. Detailed results on the HPS benchmarks are provided in Tab. 16 and 17

| Model | DPG Bench | | | | | | HPS v2.1 | HPS V3 |
| | Global | Entity | Attribute | Relation | Other | **Overall** | Overall | Overall |
|---|---|---|---|---|---|---|---|---|
| Original (50 steps) | 84.50 | 90.89 | 91.20 | 94.42 | 86.34 | 87.20 | 30.20 | 9.62 |
| ① $\tau_{\text{CA}} = \tau_{\text{DM}} \in [0, 1]$ (DMD) | 80.22 | 90.45 | 90.47 | 89.36 | 91.36 | 83.90 | 30.61 | 10.34 |
| ② $\tau_{\text{CA}} \in [0, 1], \tau_{\text{DM}} \in [0, 1]$ | 91.88 | 89.03 | 90.08 | 89.51 | 88.85 | 83.77 | 30.69 | 10.32 |
| ③ $\tau_{\text{CA}} > t, \tau_{\text{DM}} > t$ | **93.47** | 90.18 | 90.94 | 89.37 | 90.71 | 85.64 | 31.71 | 11.08 |
| ④ $\tau_{\text{CA}} > t, \tau_{\text{DM}} \in [0, 1]$ | 91.40 | **91.62** | **91.18** | 91.93 | 91.98 | **85.85** | **32.29** | **11.59** |

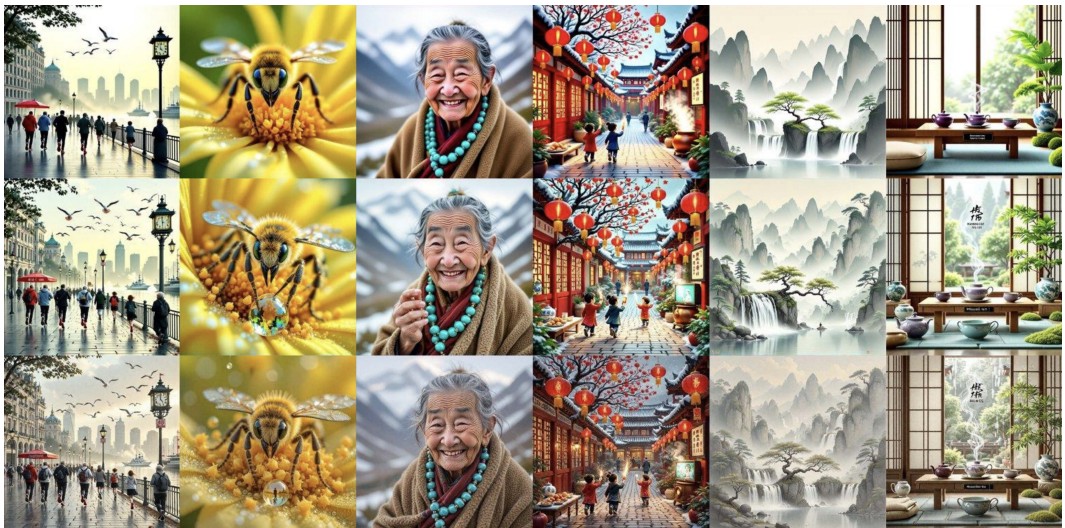

Figure 5: Un-cherry-picked qualitative comparison of different re-noising schedule configurations. Top row: ② Decoupled-Full, $\tau_{\text{CA}}, \tau_{\text{DM}} \in [0, 1]$. Middle row: ③ Coupled-Constrained, $\tau_{\text{CA}}, \tau_{\text{DM}} > t$. Bottom row: ④ our proposed Decoupled-Hybrid, $\tau_{\text{CA}} > t, \tau_{\text{DM}} \in [0, 1]$.

This understanding also clarifies the role of the re-noising timestep $\tau$ within the DM regularizer: it controls the **scope of correction**. A large $\tau$ (cleaner image) allows the real and fake scores to diverge primarily on subtle, high-frequency details. In contrast, a small $\tau$ (noisier image) destroys most details, forcing the scores to diverge on more fundamental, low-frequency elements like composition and color. This gives the DM regularizer an opportunity to correct global issues.

This leads to our final hypothesis regarding the optimal renoising schedule for DM in a few-step setting. Even late-stage generation steps, which primarily add high-frequency details, can still suffer from low-frequency artifacts like color oversaturation, either inherited from previous steps or induced by an imperfect CA schedule. To address these global issues, the DM regularizer requires a global perspective. Therefore, we propose that **the optimal DM schedule is different from that of CA: DM should function as a comprehensive regularizer, always spanning the full noise range ($\tau_{\mathbf{DM}} \in [0, 1]$), irrespective of the generator's current timestep** $t$.

### 4.3 VALIDATING THE DECOUPLED SCHEDULE HYPOTHESIS

Our mechanistic analysis in Sec.4.1 and Sec.4.2 has led to a central hypothesis: for optimal few-step distillation, the CA engine and the DM regularizer require distinct, decoupled re-noising schedules. Specifically, we proposed that the CA schedule should be constrained ($\tau_{\text{CA}} > t$) to act as a focused engine, while the DM schedule should remain global ($\tau_{\text{DM}} \in [0, 1]$) to serve as a comprehensive regularizer. In this section, we empirically validate this proposal.

To facilitate this investigation, we first generalize the DMD gradient (Eq.6) to a $\tau$-decoupled form. This allows us to assign independent re-noising schedules, $\tau_{\text{CA}}$ and $\tau_{\text{DM}}$, to the CA and DM components, respectively. The resulting "decoupled DMD" (d-DMD) gradient is formulated as:

$$\nabla_\theta \mathcal{L}_{\text{d-DMD}} = \mathbb{E}\left[ -\left( \left( s_{\text{cond}}^{\text{real}}(\mathbf{x}_{\tau_{\text{DM}}}) - s_{\text{cond}}^{\text{fake}}(\mathbf{x}_{\tau_{\text{DM}}}) \right) + (\alpha - 1)\left( s_{\text{cond}}^{\text{real}}(\mathbf{x}_{\tau_{\text{CA}}}) - s_{\text{uncond}}^{\text{real}}(\mathbf{x}_{\tau_{\text{CA}}}) \right) \right) \frac{\partial G_\theta(z_t)}{\partial \theta} \right], \quad (8)$$

Table 2: Comparison of 4-step SDXL distillation. Our 'Decoupled' method strictly follows the DMD2 training setting but employs our proposed decoupled-hybrid (④) re-noising schedule. Indicators are evaluated on 10k COCO2014-val prompts.

| Method | FID↓ | CLIP-S ↑ | ImageReward ↑ | HPS V2.1 ↑ | HPS V3 ↑ |
|---|---|---|---|---|---|
| LCM (Luo et al., 2023a) | 22.27 | 31.71 | 39.56 | 28.00 | 6.45 |
| Turbo (Sauer et al., 2024b) | 27.27 | 32.16 | 46.09 | 29.83 | 9.09 |
| Lightning (Lin et al., 2024) | 24.49 | 32.31 | 57.48 | 30.30 | 9.48 |
| Flash (Chadebec et al., 2025) | 22.96 | 31.84 | 19.04 | 27.71 | 6.49 |
| PCM (Wang et al., 2024) | 24.13 | 32.52 | 64.73 | **30.76** | 9.46 |
| DMD2 (Yin et al., 2024a) | 18.95 | 33.14 | 71.01 | 30.64 | 9.64 |
| Decoupled (Ours) | **17.80** | **33.62** | **78.61** | 30.34 | **9.79** |

where d-DMD is short for decoupled DMD. This modification allows us to decouple the renoising schedule of DM and CA, allowing principled experimental analysis. With this formulation, we design an ablation study to evaluate four distinct schedule configurations for a 4-step generator:

① **Coupled-Shared:** The original DMD approach where $\tau_{CA} = \tau_{DM}$, sampled from $[0, 1]$.

② **Decoupled-Full:** Both schedules are independent but cover the full range, $\tau_{CA}, \tau_{DM} \in [0, 1]$.

③ **Decoupled-Constrained:** Both schedules are independent and constrained, $\tau_{CA}, \tau_{DM} > t$.

④ **Decoupled-Hybrid:** The engine is constrained while the regularizer is not, $\tau_{CA} > t, \tau_{DM} \in [0, 1]$.

The results, presented in Tab. 1 for the Lumina-Image-2.0 model (Qin et al., 2025), provide strong evidence for our hypothesis. First, we confirm that merely decoupling the schedules while keeping them global ② yields negligible impact compared to the baseline ①, demonstrating that the benefit comes from the schedule's range, not just its independence. More importantly, both configurations with constrained schedules (③ and ④) significantly outperform the baselines across multiple benchmarks (Hu et al., 2024b; Wu et al., 2023; Ma et al., 2025). Crucially, our proposed Decoupled-Hybrid setting ④ consistently achieves the best overall scores, validating our core proposal.

The qualitative results in Fig. 5 offer further visual confirmation. Compared to the global schedule (②, top row), constraining the CA engine (③, middle row) introduces richer, finer details, confirming the benefit of a focused engine. However, this configuration still suffers from color oversaturation, a low-frequency artifact that its constrained DM regularizer fails to correct. In stark contrast, our Decoupled-Hybrid setting (④, bottom row) retains these enhanced details while effectively mitigating the saturation artifacts, yielding the most visually appealing and natural-looking images. These observations are decisively corroborated by a comprehensive **user study** (Sec. D), where model ④ achieved a unanimous 100% preference rate in model-level comparisons. 15 annotators consistently justified their choice by its ability to generate richer details, a more realistic and less "greasy" appearance, and fewer structural deformities. Furthermore, in a three-way image-level ranking, model ④ was ranked first in 59.8% of cases, significantly outperforming the next-best model (③ at 33.8%).

Experiments on SDXL (Tab. 2) situate our findings within the broader landscape. For a rigorous comparison with DMD2 (Yin et al., 2024a), we adopted their exact training configuration, including the GAN loss, and only replaced their re-noising schedule with our Decoupled-Hybrid approach. The results demonstrate a clear advantage, confirming the effectiveness of our proposed schedule.

Finally, to explore the full potential of our algorithm on top-tier models, we distilled a high-performance internal 6B model (Fig. 7). The resulting 4-step generator achieves performance largely on par with the original 80-NFE teacher (Tab. 6-15) at 95% NFE reduction, and exhibits superior fine details and texture compared to the conventional DMD loss (①) as shown in Fig. 6.

In summary, these results strongly validate that our engine-regularizer decomposition not only provides a deeper understanding but also unlocks tangible performance improvements, demonstrating the practical value of our new perspective.

## 5 CONCLUSION AND LIMITATIONS

In this work, we challenge the conventional understanding of DMD practice in complex tasks like text-to-image, revealing a functional decoupling with **CFG Augmentation (CA)** as the primary

engine for few-step conversion and **Distribution Matching (DM)** as the regularizer. This new perspective allowed us to observe the distinct properties of each component and propose a principled improvement–a decoupled re-noising schedule.

However, a fundamental question remains unanswered: why does CA possess such a remarkable ability to convert a diffusion model into a few-step generator? We find that providing a precise answer is highly challenging, partly because the mechanism of CFG itself remains largely enigmatic. For interested readers, we share our high-level, preliminary understanding and explanation of this issue in Sec. B. Nevertheless, we acknowledge that a significant gap remains towards a rigorously accurate explanation, and we intend to explore this topic further in our future work.

## 6 ACKNOWLEDGMENT

This study was supported in part by the National Natural Science Foundation of China (No. 62206272), in part by National Key R&D Program of China Project 2022ZD0161100, in part by the Centre for Perceptual and Interactive Intelligence, a CUHK-led InnoCentre under the InnoHK initiative of the Innovation and Technology Commission of the Hong Kong Special Administrative Region Government, in part by NSFC-RGC Project N_CUHK498/24, and in part by Guangdong Basic and Applied Basic Research Foundation (No. 2023B1515130008, XW).

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

## A    DECLARATION ON THE USE OF AI TOOLS

In the research and writing process of this work, we utilized Large Language Models (LLMs) as an assistive tool. The application of LLMs was primarily limited to two aspects: 1) For writing, we used LLMs to polish language, improve expression, and obtain suggestions regarding structure and flow. 2) For coding, LLMs were used to provide debugging assistance and to generate utility scripts, such as for data visualization. We hereby declare that the core ideas, novel methodology, experimental design, and main codebase of this work were completed by the human authors. LLMs did not contribute to these core intellectual aspects.

## B    DISCUSSION: WHY DOES CA WORK?

In this section, we attempt to build a conceptual bridge to understand the efficacy of CFG Augmentation (CA) as the "engine" of few-step distillation. We do so by drawing a parallel between diffusion models and Large Language Models (LLMs) under a unified view of sequential generation.

### B.1    A PARALLEL PROBLEM: WHY CAN'T LLMs PERFORM N-TOKEN PREDICTION?

We begin by considering a parallel question in the domain of LLMs: why must they generate text token by token, rather than predicting the next N tokens simultaneously? Consider the prompt, "The richest person in the world is". Both "Elon Musk" and "Bill Gates" are plausible completions. A model cannot simultaneously predict the first and second tokens, because the choice of the second token (e.g., "Musk" or "Gates") is strictly conditional on the choice of the first ("Elon" or "Bill"). An attempt to predict both at once would risk generating incoherent combinations like "Elon Gates" or "Bill Musk", or more likely, an averaged and meaningless representation.

The fundamental reason for this limitation is that the model's role is to predict a probability distribution for the next token, $P(\text{token}_1|\text{prompt})$. It cannot, however, control the outcome of the probabilistic sampling process that selects a single token from this distribution. This external, uncontrollable sampling event breaks the model's predictive flow. No matter how powerful the model, it cannot bypass this external intervention to predict $\text{token}_2$, because any prediction it makes could conflict with the yet-undetermined outcome of $\text{token}_1$.

### B.2    A GENERAL FRAMEWORK FOR SEQUENTIAL GENERATION

We can abstract this to a higher level. Any sequential generator, at each step, faces three types of information:

- **Type 1 (Determined):** Information that is already fixed and known (e.g., the input prompt).
- **Type 2 (Directly Determinable):** Information for which the model can predict a distribution of possibilities for the very next step.
- **Type 3 (Directly Undeterminable):** Information that can only be determined after some Type 2 information is resolved and becomes Type 1.

The process of sequential generation is the iterative conversion of information from Type 2 to Type 1, which in turn allows Type 3 to become Type 2. Importantly, we claim that **the existence of Type 3 information, whose resolution is contingent on an uncontrollable external decision on Type 2 information, is the core reason for iterative generation**.

### B.3    FROM UNCONTROLLABLE INTERVENTION TO A DETERMINISTIC PATTERN

This analysis also points to a potential strategy for converting a sequential generator into a one-step generator: if the random, external decision-making process could be replaced with a **deterministic decision pattern**, and this pattern could be "baked into" the generator itself. For Type 2 information, what was previously a random variable with non-zero entropy would become a determinable value. This would collapse the entire decision tree into a single, predictable path, allowing all information to be resolved in one go.

### B.4 CONNECTING BACK TO DIFFUSION MODELS AND CFG AUGMENTATION

We posit that diffusion models are also sequential generators, which **first** establish low-frequency global composition (e.g., the object is a cat, not a dog) **before** adding high-frequency details (e.g., the texture of the fur). The relationship between composition and detail mirrors that of "Elon/Bill" and "Musk/Gates". We note that this viewpoint has been formally established Dieleman (2024).

Crucially, we argue that **Classifier-Free Guidance (CFG) acts as an external intervention analogous to probabilistic sampling**. While CFG is a deterministic bias, not a stochastic process, it is equally unpredictable from the model's perspective: The model is trained without awareness of CFG, and at inference, it cannot control the negative prompt or guidance scale ($\alpha$) that will be applied. Furthermore, CFG, like sampling, transforms the model's prediction from an *averaged expectation* into a specific, shifted *value*.

Our central hypothesis is this: **CFG represents a specific, deterministic decision pattern.** The CA term in the DMD objective is the mechanism that **"bakes" this decision pattern into the student generator's predictions**. By doing so, it transforms the uncontrollable external force of CFG into an internalized, predictable behavior. The generation process, which was a tree of possibilities, collapses into a single, direct path.

Returning to our LLM example, what CFG Augmentation does is akin to telling the model: "Given the current input, the external process will always choose 'Elon' as the first token. Therefore, you can safely assume the first token is 'Elon' and directly predict 'Musk'." This is, we believe, the source of CA's power in enabling few-step image generation.

We acknowledge that the preceding discussion remains at the level of high-level ideas and that our hypothesis—that "CFG represents a specific, deterministic decision pattern"—is a strong assumption. We share this perspective here primarily to stimulate further investigation into this fundamental question and to provide a potential reference point for future work. We also intend to conduct a more in-depth study of this problem in our future research.

## C PSEUDO-CODE

---

**Algorithm 1** Original&Decoupled DMD Training Procedure

---

**Require:** Pre-trained teacher model $s_{\text{real}}$, CFG scale $\alpha$, number of steps $N$, proxy loss weight $\lambda$
**Ensure:** Trained few-step generator $G_\theta$
 1: ▷ *Initialize student generator and fake model from the teacher*
 2: $G_\theta \leftarrow s_{\text{real}}$
 3: $s_{\text{fake}} \leftarrow s_{\text{real}}$
 4: **while** not converged **do**
 5:     ▷ *— Generator Update Step —*
 6:     Sample a generation step $t$ from the few-step schedule $\{t_1, \ldots, t_N\}$
 7:     Prepare generator input $z_t$ (e.g., via backward simulation for $t > t_1$)
 8:     Generate an image: $x_{\text{gen}} \leftarrow G_\theta(z_t)$
 9:     **if** 'decoupled_schedule' **then**
10:         ▷ *— Decoupled DMD behavior —*
11:         Sample CFG augmentation noise level $\tau_{\text{CA}} \sim \mathcal{U}(t, 1)$
12:         Sample Distribution Matching noise level $\tau_{\text{DM}} \sim \mathcal{U}(0, 1)$
13:         Re-noise the generated image for both schedules:
14:         $x_{\tau_{\text{CA}}} \leftarrow \text{renoise}(x_{\text{gen}}, \tau_{\text{CA}})$
15:         $x_{\tau_{\text{DM}}} \leftarrow \text{renoise}(x_{\text{gen}}, \tau_{\text{DM}})$
16:         **Withwith** torch.no_grad():
17:             ▷ *Calculate scores for the Distribution Matching (DM) term*
18:             $s_{\text{cond, DM}}^{\text{real}} \leftarrow s_{\text{real}}(x_{\tau_{\text{DM}}}, \tau_{\text{DM}}, \text{text})$
19:             $s_{\text{cond, DM}}^{\text{fake}} \leftarrow s_{\text{fake}}(x_{\tau_{\text{DM}}}, \tau_{\text{DM}}, \text{text})$
20:             ▷ *Calculate scores for the CFG Augmentation (CA) term*
21:             $s_{\text{cond, CA}}^{\text{real}} \leftarrow s_{\text{real}}(x_{\tau_{\text{CA}}}, \tau_{\text{CA}}, \text{text})$
22:             $s_{\text{uncond, CA}}^{\text{real}} \leftarrow s_{\text{real}}(x_{\tau_{\text{CA}}}, \tau_{\text{CA}}, \text{``''})$
23:         **EndWith**
24:         ▷ *Compute the two components of the update direction*
25:         $\Delta_{\text{DM}} \leftarrow s_{\text{cond, DM}}^{\text{real}} - s_{\text{cond, DM}}^{\text{fake}}$
26:         $\Delta_{\text{CA}} \leftarrow (\alpha - 1)\left(s_{\text{cond, CA}}^{\text{real}} - s_{\text{uncond, CA}}^{\text{real}}\right)$
27:         $\Delta_{\text{total}} \leftarrow \Delta_{\text{DM}} + \Delta_{\text{CA}}$
28:     **else**
29:         ▷ *— Original DMD behavior —*
30:         Sample a single noise level $\tau \sim \mathcal{U}(0, 1)$
31:         Re-noise the generated image: $x_\tau \leftarrow \text{renoise}(x_{\text{gen}}, \tau)$
32:         **Withwith** torch.no_grad():
33:             $s_{\text{cond}}^{\text{real}} \leftarrow s_{\text{real}}(x_\tau, \tau, \text{text})$
34:             $s_{\text{uncond}}^{\text{real}} \leftarrow s_{\text{real}}(x_\tau, \tau, \text{``''})$
35:             $s_{\text{cond}}^{\text{fake}} \leftarrow s_{\text{fake}}(x_\tau, \tau, \text{text})$
36:             $s_{\text{cfg}}^{\text{real}} \leftarrow s_{\text{uncond}}^{\text{real}} + \alpha(s_{\text{cond}}^{\text{real}} - s_{\text{uncond}}^{\text{real}})$
37:         **EndWith**
38:         ▷ *Compute the combined update direction*
39:         $\Delta_{\text{total}} \leftarrow s_{\text{cfg}}^{\text{real}} - s_{\text{cond}}^{\text{fake}}$
40:     **end if**
41:     ▷ *Update generator by minimizing the proxy loss*
42:     $\mathcal{L}_{\text{proxy}} \leftarrow ||G_\theta(z_t) - \text{stop\_grad}(G_\theta(z_t) + \lambda\Delta_{\text{total}})||^2$
43:     Update $G_\theta$ by minimizing $\mathcal{L}_{\text{proxy}}$
44:
45:     ▷ *— Fake Model Update Step —*
46:     ▷ *This step can be run multiple times per generator update (TTUR)*
47:     Sample a new noise level $\tau' \sim \mathcal{U}(0, 1)$
48:     Generate a new image with detached gradient: $x'_{\text{gen}} \leftarrow \text{stop\_grad}(G_\theta(z_t))$
49:     Re-noise the new image: $x'_{\tau'} \leftarrow \text{renoise}(x'_{\text{gen}}, \tau')$
50:     $\mathcal{L}_{\text{denoise}} \leftarrow ||s_{\text{fake}}(x'_{\tau'}, \tau') - x'_{\text{gen}}||^2$
51:     Update $s_{\text{fake}}$ using $\nabla\mathcal{L}_{\text{denoise}}$
52: **end while**

---

# D   USER STUDY

To further validate the effectiveness of our proposed Decoupled-Hybrid schedule (④), we conducted a comprehensive user study comparing the four few-step models from the ablation in Table 1. The study was divided into two parts: a per-image ranking evaluation and a per-model side-by-side comparison.

## D.1   PER-IMAGE RANKING EVALUATION

In this part, we compared the visual quality of models ② (Decoupled-Full), ③ (Decoupled-Constrained), and ④ (Decoupled-Hybrid). We randomly selected 500 prompts from the HPSv2 benchmark Wu et al. (2023) and generated one image for each prompt using the three models. We then invited 10 professional annotators to perform a forced ranking of the three images in each set based on their overall quality (e.g., a ranking of 2>3>1, with ties disallowed). To prevent positional bias, the order of the three images within each triplet was randomized, so annotators could not determine which model generated which image based on its position.

The quantitative results are presented in Table 3 and Table 4. As shown, our proposed Decoupled-Hybrid schedule (④) demonstrates a significant and consistent advantage over the other configurations. It achieved a first-place ranking in **59.8%** of the evaluations, far surpassing the 33.8% of the next-best model, ③. The pairwise comparison in Table 4 further confirms this superiority, where model ④ achieved win rates of 60.6% and 83.4% against models ③ and ②, respectively.

Table 3: Overall performance and rank distribution from the per-image user study. Our Decoupled-Hybrid schedule (④) was ranked first in the majority of evaluations.

| Model | Avg. Rank ↓ | 1st Place (%) ↑ | 2nd Place (%) | 3rd Place (%) |
|---|---|---|---|---|
| ② Decoupled-Full | 2.748 | 6.4 | 12.4 | **81.2** |
| ③ Decoupled-Constrained | 1.692 | 33.8 | 63.2 | 3.0 |
| ④ Decoupled-Hybrid | **1.560** | **59.8** | 24.4 | 15.8 |

Table 4: Pairwise win rates (%) from the per-image ranking evaluation. Each cell (row, col) shows the percentage of times the model in the row was ranked higher than the model in the column.

| | ② Decoupled-Full | ③ Decoupled-Constrained | ④ Decoupled-Hybrid |
|---|---|---|---|
| ② Decoupled-Full | – | 8.6 | 16.6 |
| ③ Decoupled-Constrained | 91.4 | – | 39.4 |
| ④ Decoupled-Hybrid | **83.4** | **60.6** | – |

## D.2   PER-MODEL SIDE-BY-SIDE COMPARISON

We further conducted a per-model comparison to gauge the overall preference between different schedules. In this setup, we performed three separate side-by-side comparisons: ① (Coupled-Shared) vs. ④, ② (Decoupled-Full) vs. ④, and ③ (Decoupled-Constrained) vs. ④. For each comparison, we randomly sampled 200 prompts from the HPSv2 benchmark (using a different random seed for each pair) and displayed the generated images in a fixed two-column layout. We then asked 15 annotators to review all 200 image pairs and make a single, model-level judgment on which model (left or right) performed better overall. They were also asked to provide a brief justification for their choice.

Table 5: Quantitative results from the per-model, side-by-side user study with 15 annotators. Our Decoupled-Hybrid model (④) achieved a unanimous 100% preference rate in all comparisons.

| Comparison | # Prompts | Preference for Model ④ (%) |
|---|---|---|
| ① (Coupled-Shared) vs. ④ | 200 | **100%** |
| ② (Decoupled-Full) vs. ④ | 200 | **100%** |
| ③ (Decoupled-Constrained) vs. ④ | 200 | **100%** |

The quantitative results, summarized in Table 5, show a decisive victory for our proposed Decoupled-Hybrid model (④). It was unanimously preferred in all three head-to-head comparisons, achieving a **100%** win rate.

The justifications provided by the annotators shed light on the reasons for this strong preference. The most frequently cited advantages of our model (④) were its ability to generate **richer details**, produce a more **realistic/less over-saturated/not greasy** texture/coloring, and exhibit **fewer anatomical or structural deformities**. **The two-column comparison tables presented to the annotators for the per-model evaluation, along with the complete textual justifications from annotators, are included in the supplementary material zip file for reference.**

# E    DISTILLATION OF A HIGH-PERFORMANCE 6B INTERNAL MODEL

## E.1    QUALITATIVE VISUALIZATION

To fully unleash the potential of our algorithm and explore its performance on high-performance models, we also conduct experiments on an internal flow-matching-based 6B model.

Fig. 6 presents a qualitative comparison between our proposed decoupled-hybrid schedule (④) and the conventional DMD loss (①). Our method markedly leads to a pronounced improvement in the articulation of fine details and the authenticity of surface textures.

Fig. 7 showcases the performance of our distilled 4-step model, which demonstrates strong capabilities in text rendering, artistic versatility, and photorealistic quality.

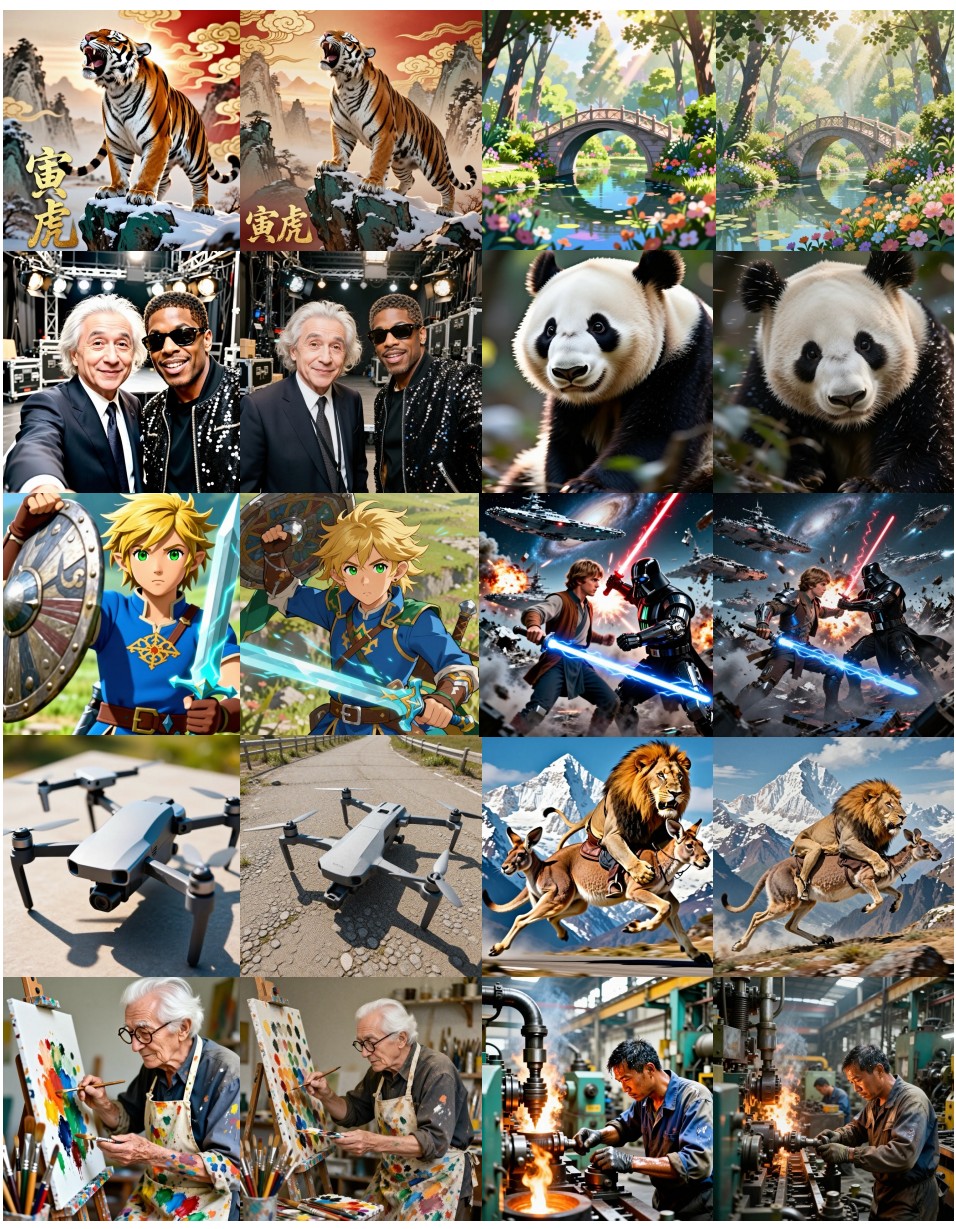

Figure 6: Qualitative comparison on distilling our internal 6B high-performance model to 4 step. For a pair of images, the left is generated by the model trained with conventional DMD loss, while the right is trained with our proposed decoupled-hybrid schedule

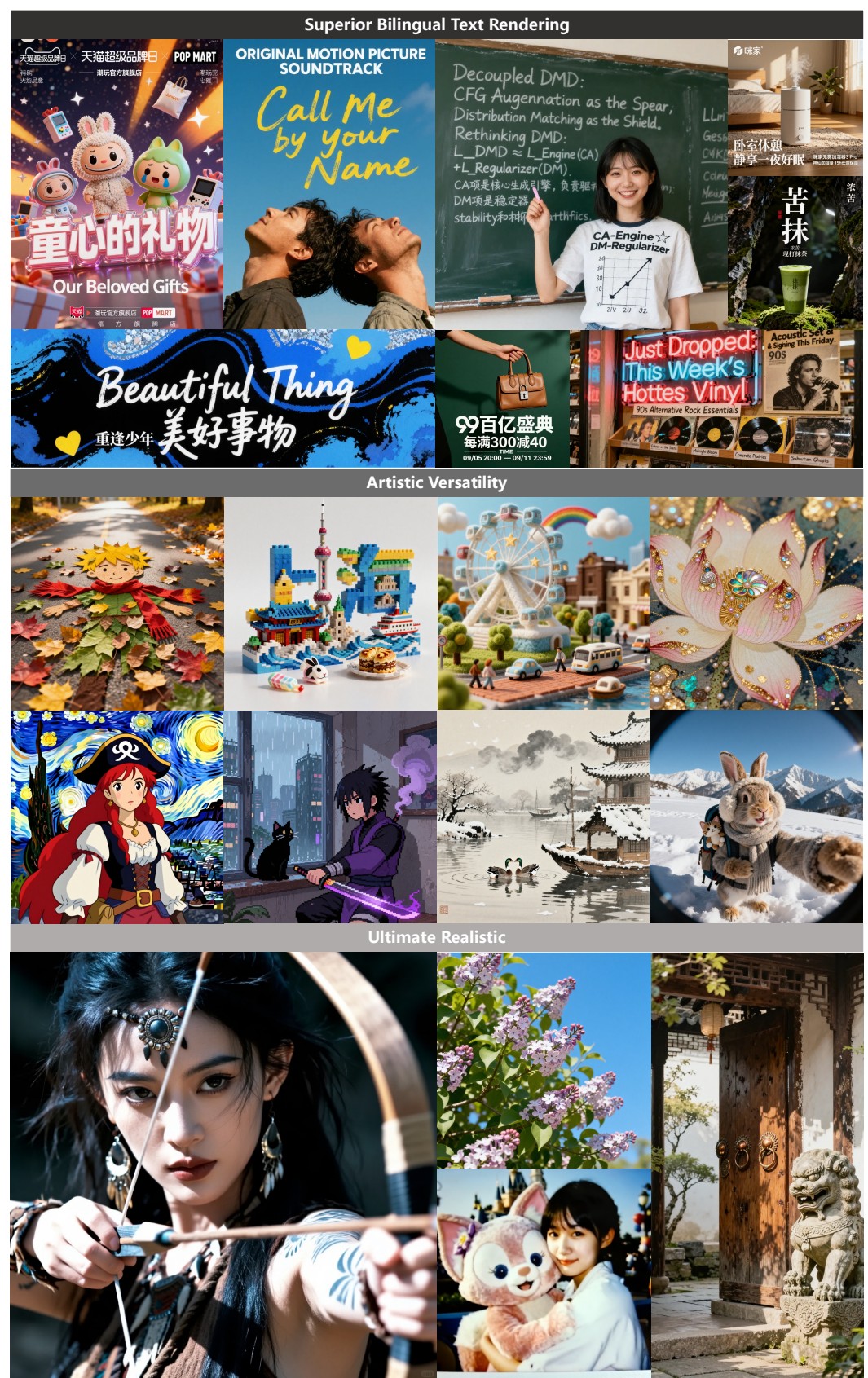

Figure 7: Performance demonstration of our 4-step distilled internal 6B model

## E.2 QUANTITATIVE COMPARISON

Tab. 6 through Tab. 15 provide comprehensive quantitative comparisons of the model's performance on various benchmarks before and after distillation. The original 6B model requires 40 steps with CFG for inference, totaling 80 NFE (Number of Function Evaluations). After distillation with our decoupled-hybrid schedule, the resulting 4-step model achieves performance that is largely on par with the original teacher at 95% NFE reduction, with only minor fluctuations across different metrics.

Table 6: Summary of Quantitative Evaluation Results Across All Benchmarks.

| Model | DPG_Bench | GenEval | OneIG | | TIIF-Bench | | PRISM-Bench | | CVTG-2K | LongTextBench | |
|---|---|---|---|---|---|---|---|---|---|---|---|
| | | | EN | ZH | SHORT | LONG | EN | ZH | | EN | ZH |
| SDXL Podell et al. (2023) | 74.65 | - | 0.316 | - | - | - | 57.0 | - | - | - | - |
| SD3 Medium Esser et al. (2024) | 84.08 | 0.62 | - | - | 67.46 | 66.09 | 66.1 | - | - | - | - |
| FLUX.1 [Dev] Labs (2024) | 83.84 | 0.66 | 0.434 | - | 71.09 | 71.78 | 68.5 | - | 0.4965 | 0.607 | 0.005 |
| Lumina-Image 2.0 Qin et al. (2025) | 87.20 | 0.73 | 0.353 | 0.334 | - | - | - | - | - | - | - |
| Seedream3.0 Gao et al. (2025) | 88.27 | 0.84 | 0.530 | 0.528 | 86.02 | 84.31 | 76.2 | 74.7 | 0.5924 | 0.896 | 0.878 |
| GPT Image 1 [High] OpenAI (2025) | 85.15 | 0.84 | 0.533 | 0.474 | 89.15 | 88.29 | 80.7 | 77.7 | 0.8569 | 0.956 | 0.619 |
| Qwen-Image Wu et al. (2025) | 88.32 | 0.87 | 0.539 | 0.548 | 86.14 | 86.83 | 74.1 | 70.3 | 0.8288 | 0.943 | 0.946 |
| **Internal-6B** | 88.14 | 0.80 | **0.546** | 0.529 | 80.01 | 80.20 | 75.1 | 73.3 | 0.8604 | 0.922 | 0.945 |
| **Internal-6B-Turbo** | **88.53** | 0.78 | 0.544 | **0.531** | 82.15 | 78.94 | **75.3** | 75.3 | 0.8327 | 0.895 | **0.953** |
| **Difference** | +0.39 | -0.02 | -0.002 | +0.002 | +2.14 | -1.26 | +0.20 | +2.00 | -0.03 | -0.03 | +0.01 |

Table 7: Quantitative Evaluation Results on DPG Bench Hu et al. (2024a).

| Model | Global | Entity | Attribute | Relation | Other | Overall |
|---|---|---|---|---|---|---|
| SDXL Podell et al. (2023) | 83.27 | 82.43 | 80.91 | 86.76 | 80.41 | 74.65 |
| SD3 Medium Esser et al. (2024) | 87.90 | 91.01 | 88.83 | 80.70 | 88.68 | 84.08 |
| FLUX.1 [Dev] Labs (2024) | 74.35 | 90.00 | 88.96 | 90.87 | 88.33 | 83.84 |
| Lumina-Image 2.0 Qin et al. (2025) | - | 91.97 | 90.20 | **94.85** | - | 87.20 |
| Seedream3.0 Gao et al. (2025) | 94.31 | 92.65 | 91.36 | 92.78 | 88.24 | 88.27 |
| GPT Image 1 [High] OpenAI (2025) | 88.89 | 88.94 | 89.84 | 92.63 | 90.96 | 85.15 |
| Qwen-Image Wu et al. (2025) | 91.32 | 91.56 | 92.02 | 94.31 | **92.73** | 88.32 |
| **Interal-6B** | 93.39 | 91.22 | **93.16** | 92.22 | 91.52 | 88.14 |
| **Interal-6B-Turbo** | **94.70** | **93.24** | 92.78 | 92.01 | 92.58 | **88.53** |

Table 8: Quantitative Evaluation Results on GenEval Ghosh et al. (2023).

| Model | Single Object | Two Object | Counting | Colors | Position | Attribute Binding | Overall |
|---|---|---|---|---|---|---|---|
| SD3 Medium Esser et al. (2024) | 0.98 | 0.74 | 0.63 | 0.67 | 0.34 | 0.36 | 0.62 |
| FLUX.1 [Dev] Labs (2024) | 0.98 | 0.81 | 0.74 | 0.79 | 0.22 | 0.45 | 0.66 |
| Lumina-Image 2.0 Qin et al. (2025) | - | 0.87 | 0.67 | - | - | 0.62 | 0.73 |
| Seedream3.0 Gao et al. (2025) | 0.99 | **0.96** | **0.91** | **0.93** | 0.47 | **0.80** | 0.84 |
| GPT Image 1 [High] OpenAI (2025) | 0.99 | 0.92 | 0.85 | 0.92 | 0.75 | 0.61 | 0.84 |
| Qwen-Image Wu et al. (2025) | 0.99 | 0.92 | 0.89 | 0.88 | **0.76** | 0.77 | **0.87** |
| **Internal-6B** | **1.00** | 0.94 | 0.70 | 0.90 | 0.51 | 0.72 | 0.80 |
| **Internal-6B-Turbo** | **1.00** | 0.94 | 0.60 | 0.88 | 0.55 | 0.72 | 0.78 |

Table 9: Quantitative Evaluation Results on OneIG-EN Chang et al. (2025).

| Model | Alignment | Text | Reasoning | Style | Diversity | Overall |
|---|---|---|---|---|---|---|
| SDXL Podell et al. (2023) | 0.688 | 0.029 | 0.237 | 0.332 | **0.296** | 0.316 |
| FLUX.1 [Dev] Labs (2024) | 0.786 | 0.523 | 0.253 | 0.368 | 0.238 | 0.434 |
| Lumina-Image 2.0 Qin et al. (2025) | 0.819 | 0.106 | 0.270 | 0.354 | 0.216 | 0.353 |
| Seedream3.0 Gao et al. (2025) | 0.818 | 0.865 | 0.275 | 0.413 | 0.277 | 0.530 |
| GPT Image 1 [High] OpenAI (2025) | 0.851 | 0.857 | **0.345** | **0.462** | 0.151 | 0.533 |
| Qwen-Image Wu et al. (2025) | **0.882** | 0.891 | 0.306 | 0.418 | 0.197 | 0.539 |
| **Internal-6B** | 0.881 | **0.987** | 0.280 | 0.387 | 0.194 | **0.546** |
| **Internal-6B-Turbo** | 0.876 | 0.972 | 0.283 | 0.404 | 0.186 | 0.544 |

Table 10: Quantitative Evaluation Results on OneIG-ZH Chang et al. (2025).

| Model | Alignment | Text | Reasoning | Style | Diversity | Overall |
|---|---|---|---|---|---|---|
| Lumina-Image 2.0 Qin et al. (2025) | 0.731 | 0.136 | 0.221 | 0.343 | 0.240 | 0.334 |
| Seedream3.0 Gao et al. (2025) | 0.793 | 0.928 | 0.281 | 0.397 | 0.243 | 0.528 |
| GPT Image 1 [High] OpenAI (2025) | 0.812 | 0.650 | **0.300** | **0.449** | 0.159 | 0.474 |
| Qwen-Image Wu et al. (2025) | 0.825 | 0.963 | 0.267 | 0.405 | **0.279** | **0.548** |
| **Internal-6B** | 0.829 | **0.990** | 0.269 | 0.377 | 0.179 | 0.529 |
| **Internal-6B-Turbo** | **0.831** | 0.986 | 0.272 | 0.388 | 0.175 | 0.531 |

Table 11: Quantitative Evaluation Results on TIIF-Bench Wei et al. (2025).

| Model | Overall | | Basic_Following | | | | | | | | | | Advanced_Following | | | | | | | | | | | | Designer | | | | | | |
|---|---|---|---|---|---|---|---|---|---|---|---|---|---|---|---|---|---|---|---|---|---|---|---|---|---|---|---|---|---|---|---|
| | | | | Avg | | Attribute | | Relation | | Reasoning | | | Avg | | Attribute +Relation | | Attribute +Relation | | Attribute +Relation | | | | | | Style | | Text | | Real | | |
| | short | long | short | long | short | long | short | long | short | long | | short | long | short | long | short | long | short | long | | | | | | short | long | short | long | short | long | |
| SD3 Esser et al. (2024) | 67.46 | 66.09 | 78.32 | 77.75 | 83.33 | 79.83 | 82.07 | 78.82 | 71.07 | 74.07 | | 61.46 | 59.56 | 61.07 | 64.07 | 68.84 | 70.34 | 50.96 | 57.84 | | | | | | 66.67 | 76.67 | 59.83 | 20.83 | 63.23 | 67.34 | |
| FLUX.1 [Dev] Labs (2024) | 71.09 | 71.78 | 83.12 | 78.65 | 87.05 | 83.17 | 87.25 | 80.39 | 75.01 | 72.39 | | 65.79 | 68.54 | 67.07 | 73.69 | 73.84 | 73.34 | 69.09 | 71.59 | | | | | | 66.67 | 66.67 | 43.83 | 52.83 | 70.72 | 71.47 | |
| FLUX.1 [Pro] Labs (2024) | 67.32 | 69.89 | 79.08 | 78.91 | 78.83 | 81.33 | 82.82 | 83.82 | 75.57 | 71.57 | | 61.10 | 65.37 | 62.32 | 65.57 | 69.84 | 71.47 | 65.96 | 67.72 | | | | | | 63.00 | 63.00 | 35.83 | 55.83 | 71.80 | 68.80 | |
| Seedream3.0 Gao et al. (2025) | 86.02 | 84.31 | 87.07 | 84.93 | 90.50 | 90.00 | 89.85 | 85.94 | 80.86 | 78.86 | | 79.16 | 80.60 | 79.76 | 81.82 | 77.23 | 78.85 | 75.64 | 78.64 | | | | | | 100.00 | 93.33 | 97.17 | 87.78 | 83.21 | 83.58 | |
| GPT Image 1 [High] OpenAI (2025) | 89.15 | 88.29 | 90.75 | 89.66 | 91.33 | 87.08 | 84.57 | 84.57 | 96.32 | 97.32 | | 88.55 | 88.35 | 87.07 | 89.44 | 87.22 | 83.96 | 85.59 | 83.21 | | | | | | 90.00 | 93.33 | 89.83 | 86.83 | 89.73 | 93.46 | |
| Qwen-Image Wu et al. (2025) | 86.14 | 86.83 | 86.18 | 87.22 | 90.50 | 91.50 | 88.22 | 90.78 | 79.81 | 79.38 | | 79.30 | 80.88 | 79.21 | 78.94 | 78.85 | 81.69 | 75.57 | 78.59 | | | | | | 100.00 | 100.00 | 92.76 | 89.14 | 90.30 | 91.42 | |
| **Internal-6B** | 80.01 | 80.20 | 79.13 | 81.91 | 83.50 | 83.50 | 82.32 | 84.51 | 71.56 | 77.73 | | 74.63 | 75.63 | 78.21 | 76.48 | 75.26 | 75.89 | 66.44 | 72.56 | | | | | | 93.33 | 86.67 | 87.78 | 80.09 | 81.72 | 84.33 | |
| **Internal-6B-Turbo** | 82.15 | 78.94 | 83.95 | 81.59 | 87.00 | 84.50 | 88.17 | 83.07 | 76.68 | 77.21 | | 75.23 | 74.63 | 77.74 | 76.65 | 72.02 | 72.95 | 72.03 | 72.68 | | | | | | 96.67 | 83.33 | 85.07 | 78.73 | 83.96 | 81.34 | |

Table 12: Quantitative Evaluation Results on PRISM-Bench-EN Fang et al. (2025).

| Model | Imagination | | | Entity | | | Text rendering | | | Style | | | Affection | | | Composition | | | Long text | | | Overall | | |
|---|---|---|---|---|---|---|---|---|---|---|---|---|---|---|---|---|---|---|---|---|---|---|---|---|
| | Ali. | Aes. | Avg. | Ali. | Aes. | Avg. | Ali. | Aes. | Avg. | Ali. | Aes. | Avg. | Ali. | Aes. | Avg. | Ali. | Aes. | Avg. | Ali. | Aes. | Avg. | Ali. | Aes. | Avg. |
| SDXL Podell et al. (2023) | 54.5 | 34.1 | 44.3 | 71.1 | 65.0 | 68.0 | 18.6 | 37.3 | 27.9 | 71.7 | 72.6 | 72.1 | 78.7 | 66.5 | 72.6 | 72.2 | 67.8 | 70.0 | 54.1 | 34.5 | 44.3 | 60.1 | 54.0 | 57.0 |
| SD3 Medium Esser et al. (2024) | 64.3 | 37.7 | 51.0 | 69.4 | 63.3 | 66.3 | 38.5 | 63.3 | 50.9 | 74.6 | 79.5 | 77.0 | 80.5 | 75.5 | 78.0 | 85.6 | 79.5 | 82.5 | 63.4 | 50.3 | 56.8 | 68.0 | 64.2 | 66.1 |
| FLUX.1 [Schnell] Labs (2024) | 62.8 | 35.6 | 49.2 | 64.8 | 56.8 | 60.8 | 54.3 | 68.1 | 61.2 | 70.3 | 71.5 | 70.9 | 75.4 | 65.9 | 70.6 | 81.7 | 75.6 | 78.6 | 68.7 | 54.4 | 61.5 | 68.3 | 61.1 | 64.7 |
| FLUX.1 [Dev] Labs (2024) | 65.5 | 42.9 | 54.2 | 70.6 | 61.9 | 66.2 | 52.3 | 73.0 | 62.6 | 72.6 | 74.2 | 73.4 | 86.0 | 72.9 | 79.4 | 87.4 | 75.8 | 81.6 | 70.5 | 53.8 | 62.1 | 72.1 | 64.9 | 68.5 |
| Seedream3.0 Gao et al. (2025) | 75.8 | 38.0 | 56.9 | 81.3 | 74.2 | 77.7 | 58.8 | 74.0 | 66.4 | 84.4 | 84.1 | 84.2 | 90.5 | 74.6 | 82.5 | 93.6 | 85.1 | 89.3 | 76.2 | 76.4 | 76.3 | 80.1 | 72.3 | 76.2 |
| GPT Image 1 [High] OpenAI (2025) | 79.8 | 53.3 | 66.6 | 87.3 | 81.0 | 84.1 | 66.7 | 86.8 | 76.8 | 87.3 | 87.8 | 87.5 | 88.1 | 79.8 | 84.0 | 92.2 | 84.9 | 88.5 | 77.2 | 77.5 | 77.4 | 82.7 | 78.7 | 80.7 |
| Qwen-Image Wu et al. (2025) | 75.5 | 37.4 | 56.5 | 79.5 | 64.5 | 72.0 | 57.9 | 71.2 | 64.5 | 86.6 | 84.4 | 85.5 | 89.9 | 70.4 | 80.1 | 93.9 | 79.5 | 86.7 | 76.8 | 70.9 | 73.8 | 80.0 | 68.3 | 74.1 |
| **Internal-6B** | 75.5 | 43.0 | 59.2 | 75.7 | 71.0 | 73.4 | 60.9 | 72.4 | 66.7 | 78.2 | 87.2 | 82.7 | **92.0** | **79.9** | **86.0** | 90.9 | 85.2 | 88.1 | 70.7 | 68.9 | 69.8 | 77.7 | 72.5 | 75.1 |
| **Internal-6B-Turbo** | 72.9 | 37.6 | 55.3 | 73.2 | 70.4 | 71.8 | 55.8 | 74.6 | 65.2 | 80.1 | **91.5** | 85.8 | 90.1 | **79.9** | 84.9 | 91.1 | **86.8** | 89.0 | 71.1 | **79.0** | 75.0 | 76.3 | 74.2 | 75.3 |

Table 13: Quantitative Evaluation Results on PRISM-Bench-ZH Fang et al. (2025).

| Model | Imagination | | | Entity | | | Text rendering | | | Style | | | Affection | | | Composition | | | Long text | | | Overall | | |
|---|---|---|---|---|---|---|---|---|---|---|---|---|---|---|---|---|---|---|---|---|---|---|---|---|
| | Ali. | Aes. | Avg. | Ali. | Aes. | Avg. | Ali. | Aes. | Avg. | Ali. | Aes. | Avg. | Ali. | Aes. | Avg. | Ali. | Aes. | Avg. | Ali. | Aes. | Avg. | Ali. | Aes. | Avg. |
| Seedream3.0 Gao et al. (2025) | 71.4 | 36.6 | 54.0 | 72.6 | 88.0 | 79.4 | 74.1 | 88.0 | 81.1 | **79.0** | 71.4 | 75.2 | 90.3 | 83.2 | 86.8 | 73.0 | 71.2 | 72.1 | 76.2 | 73.2 | 74.7 |
| GPT Image 1 [High] OpenAI (2025) | **73.0** | 37.6 | **55.3** | 80.4 | 82.1 | 81.3 | 73.1 | 89.9 | 81.5 | **77.1** | **92.4** | **84.8** | 78.0 | 77.8 | 77.9 | 91.9 | 85.7 | **88.8** | 72.4 | **76.3** | **74.4** | **78.0** | **77.4** | **77.7** |
| Qwen-Image Wu et al. (2025) | 71.4 | 29.9 | 50.7 | 74.7 | 67.8 | 71.3 | 64.3 | 73.1 | 68.7 | 75.2 | 83.2 | 79.2 | 77.3 | 64.5 | 70.9 | 89.8 | 74.1 | 82.0 | 72.6 | 65.8 | 69.2 | 75.0 | 65.5 | 70.3 |
| **Internal-6B** | 71.5 | **38.4** | 54.5 | 70.6 | 69.8 | 70.2 | **76.9** | 84.8 | 80.8 | 75.7 | 85.1 | 80.5 | 78.8 | 70.5 | 74.7 | 90.1 | 77.8 | 83.9 | **73.9** | 62.4 | 68.2 | 76.9 | 69.7 | 73.3 |
| **Internal-6B-Turbo** | 69.5 | 34.1 | 51.6 | 70.6 | 73.7 | 72.2 | 76.8 | **90.0** | **83.4** | 74.1 | 88.2 | 81.2 | 77.6 | 73.5 | 75.5 | 89.3 | **88.0** | 88.6 | 71.6 | 75.6 | 73.6 | 75.7 | **74.9** | 75.3 |

Table 14: Quantitative Evaluation Results on CVTG-2K Du et al. (2025).

| Model | Word Accuracy | | | | | NED | CLIPScore |
|---|---|---|---|---|---|---|---|
| | 2 regions | 3 regions | 4 regions | 5 regions | average | | |
| FLUX.1 [Dev] Labs (2024) | 0.6089 | 0.5531 | 0.4661 | 0.4316 | 0.4965 | 0.6879 | 0.7401 |
| Seedream3.0 Gao et al. (2025) | 0.6282 | 0.5962 | 0.6043 | 0.5610 | 0.5924 | 0.8537 | 0.7821 |
| GPT Image 1 [High] OpenAI (2025) | **0.8779** | 0.8659 | **0.8731** | 0.8218 | 0.8569 | **0.9478** | 0.7982 |
| Qwen-Image Wu et al. (2025) | 0.8370 | 0.8364 | 0.8313 | 0.8158 | 0.8288 | 0.9116 | 0.8017 |
| **Internal-6B** | 0.8652 | **0.8736** | 0.8664 | **0.8404** | **0.8604** | 0.9419 | 0.8107 |
| **Internal-6B-Turbo** | 0.8417 | 0.8457 | 0.8350 | 0.8148 | 0.8327 | 0.9287 | **0.8411** |

Table 15: Quantitative Evaluation Results on LongText-Bench Geng et al. (2025).

| Model | LongText-Bench-EN | LongText-Bench-ZH |
|---|---|---|
| FLUX.1 [Dev] Labs (2024) | 0.607 | 0.005 |
| Seedream3.0 Gao et al. (2025) | 0.896 | 0.878 |
| GPT Image 1 [High] OpenAI (2025) | **0.956** | 0.619 |
| Qwen-Image Wu et al. (2025) | 0.943 | 0.946 |
| **Internal-6B** | 0.922 | 0.945 |
| **Internal-6B-Turbo** | 0.895 | **0.953** |

# F ADDITIONAL EXPERIMENTAL RESULTS

## F.1 DETAILED RESULTS OF RE-NOISING SCHEDULE ABLATION

In Tab. 1 of the main text, we have already presented the overall performance comparison for different re-noising schedule configurations. In Tab. 16 and Tab. 17, we additionally provide the fine-grained results on the HPS v2.1 and HPS v3 benchmarks, respectively.

Table 16: Detailed results of the Tab. 1 experiment on the HPS v2.1 benchmark.

| Method | Concept-Art | Photo | Anime | Paintings | **Average** |
|---|---|---|---|---|---|
| Original (50 steps) | 30.35 | 28.24 | 31.74 | 30.47 | 30.20 |
| ① $\tau_{CA} = \tau_{DM} \in [0, 1]$ (DMD) | 30.31 | 29.95 | 31.72 | 30.45 | 30.61 |
| ② $\tau_{CA} \in [0, 1], \tau_{DM} \in [0, 1]$ | 30.31 | 30.25 | 31.85 | 30.34 | 30.69 |
| ③ $\tau_{CA} > t, \tau_{DM} > t$ | 31.48 | **30.87** | 32.98 | 31.51 | 31.71 |
| ④ $\tau_{CA} > t, \tau_{DM} \in [0, 1]$ | **32.37** | **30.87** | **33.61** | **32.31** | **32.29** |

Table 17: Detailed results of the Tab. 1 experiment on the HPS v3 benchmark.

| Method | Animals | Architecture | Arts | Characters | Design | Food |
|---|---|---|---|---|---|---|
| Original (50 steps) | 9.562 | 10.418 | 9.292 | 10.822 | 8.358 | 10.160 |
| ① $\tau_{CA} = \tau_{DM} \in [0, 1]$ (DMD) | 10.377 | 11.537 | 9.077 | 11.061 | 8.112 | 11.520 |
| ② $\tau_{CA} \in [0, 1], \tau_{DM} \in [0, 1]$ | 10.338 | 11.539 | 8.929 | 11.101 | 8.059 | 11.572 |
| ③ $\tau_{CA} > t, \tau_{DM} > t$ | 11.309 | 12.388 | 9.690 | 11.975 | 8.694 | 12.096 |
| ④ $\tau_{CA} > t, \tau_{DM} \in [0, 1]$ | 11.873 | 12.899 | 10.351 | 12.562 | 9.308 | 12.425 |

| Method | Nat. Sce. | Others | Plants | Products | Science | Transportation |
|---|---|---|---|---|---|---|
| Original (50 steps) | 9.781 | 9.623 | 9.881 | 9.409 | 8.477 | 9.659 |
| ① $\tau_{CA} = \tau_{DM} \in [0, 1]$ (DMD) | 10.138 | 10.797 | 10.528 | 10.994 | 8.417 | 11.503 |
| ② $\tau_{CA} \in [0, 1], \tau_{DM} \in [0, 1]$ | 10.046 | 10.787 | 10.600 | 11.049 | 8.358 | 11.484 |
| ③ $\tau_{CA} > t, \tau_{DM} > t$ | 11.177 | 11.553 | 11.390 | 11.508 | 8.905 | 12.256 |
| ④ $\tau_{CA} > t, \tau_{DM} \in [0, 1]$ | 11.592 | 12.138 | 11.846 | 11.841 | 9.451 | 12.782 |

## F.2 Qualitative Comparison of Different Regularizers

In Fig. 3 of the main text, we have already traced the quantitative indicators of combining the CFG Augmentation (CA) with different regularizers. In Fig. 8, we additionally provide a visualization of the generated samples from this experiment.

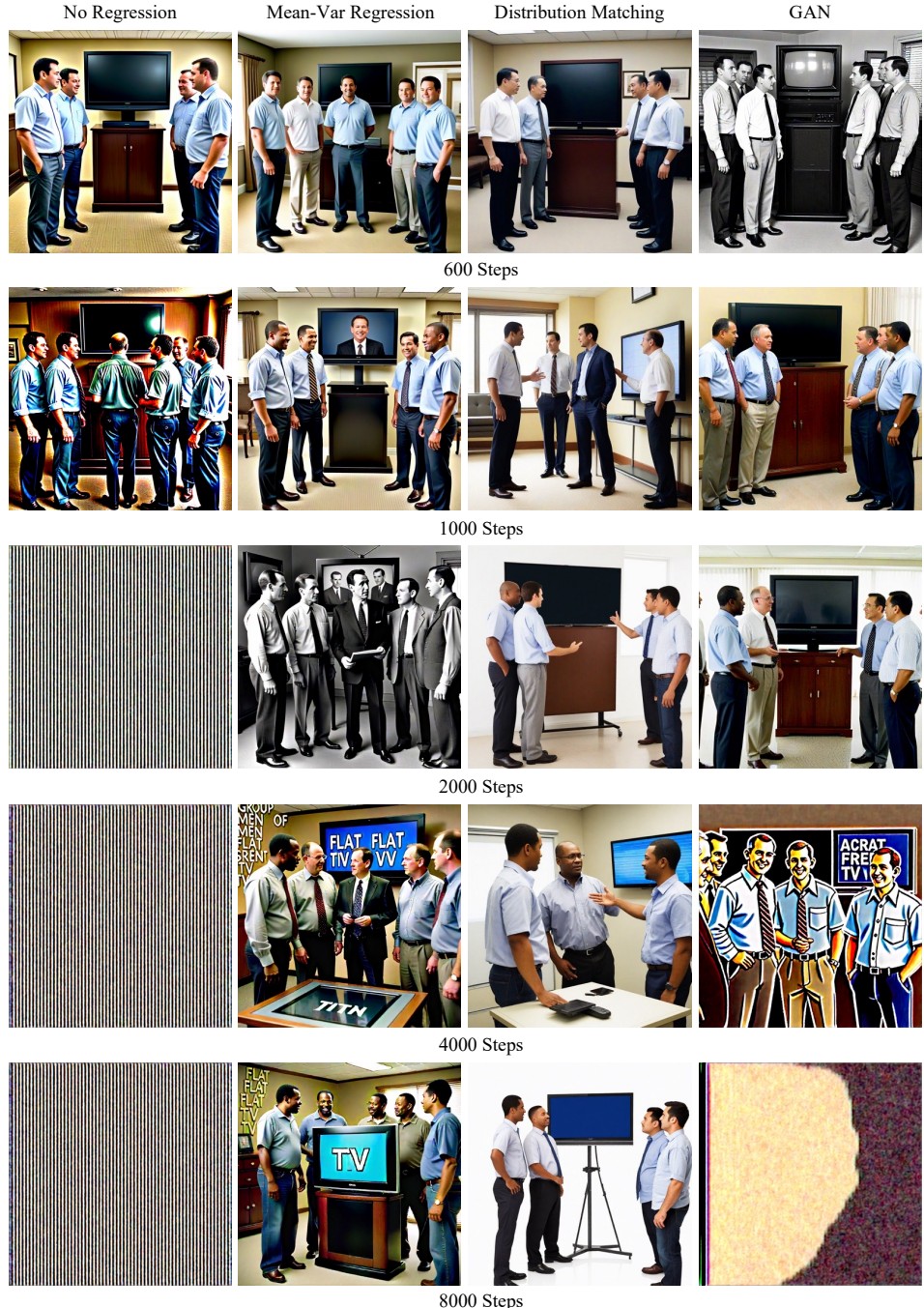

Figure 8: Sample visualization on combining training CA with different regularizers (Fig. 3)

