# OpenReview forum: "Decoupled DMD: CFG Augmentation as the Spear, Distribution Matching as the Shield"
_ICLR.cc/2026/Conference — ICLR 2026 Poster_

### Official Review · Reviewer_rsWL · 2025-10-29

**Soundness:** 3
**Presentation:** 3
**Contribution:** 2
**Rating:** 4
**Confidence:** 3

**Summary:**

This work revisits what makes DMD effective. While DMD’s success is usually credited to matching the student’s output distribution with a teacher model, the authors show that the true driver is an overlooked component called CFG Augmentation (CA). They find that CA acts as the core engine of distillation, while the Distribution Matching (DM) term mainly serves as a stabilising regularizer. They also show that recognising this separation enables a clearer understanding of DMD and allows improvements such as decoupling noise schedules for better performance.

**Strengths:**

I really like this research topic and believe the distribution-matching distillation is an under-explored topic, and only from a divergence perspective, it can't answer why it works or why it doesn't work in some scenarios, so I think the topic of this paper is very valuable.

The experiments are also sound, which can support the argument.

**Weaknesses:**

My major concern with this paper is that I found the conclusion a little bit conclusive.

The argument is CFG Augmentation is the engine for dilatation, and Distribution Matching is the regularizer for stability.

However, many CIFAR experiments don't use label-conditioned and can achieve one-step distillation, e.g. the original diff-intruct paper or more recent paper: https://arxiv.org/pdf/2502.08005. In this case, the pure driven engine is only the distribution matching term, which couldn't be explained by the hypotheses introduced in the paper.

It may be possible that CFG can play a key role in the conditional generation, but it is hard to say DM is not the engine.

Minor: The distillation also relates to the student model score estimation quality, initialisation, teacher model's score quality, etc... It would be good to add some analysis on that.

**Questions:**

See above, why the unconditional CIFAR works with only DM term?

Happy to increase the score if the concern could be solved.

---

> ### Author Response · Authors · 2025-11-13
> **Thank you for your comments!**
>
> Dear Reviewer rsWL,
>
> We sincerely thank you for your insightful review and valuable feedback. We are encouraged that you found our research topic valuable and our experiments sound. We particularly appreciate your thoughtful critique regarding our conclusion, which you rightly pointed out as potentially being "a little bit conclusive." Your comments have been instrumental in helping us refine the claims and improve the clarity of our paper.
>
> We apologize that in the original manuscript, we have implicitly concentrated the scope of consideration to complex tasks like text-to-image without explicitly clarifying this. Consequently, our earlier assertions were presented without the required constraints. In response to your feedback, we have made revisions to the manuscript:
>
> - **Refining Our Central Claim:** Following your and Reviewer uRhT's suggestions, we have revised our central claim to be more precise: **"in complex scenarios like text-to-image generation, the CA term is the primary engine for DMD"**
> - **Moderating Language:** We have adjusted statements that might have sounded overly dismissive of the DM term. For instance, phrase "$\mathcal{L}_{DMD\text{-theory}}$​ usually leads to poor performance in practice" has been revised to "**While $\mathcal{L}_{DMD\text{-theory}}$​ proved to be effective on simple generation tasks like CIFAR, its performance on complex tasks like text-to-image is usually unsatisfactory.**"
>
> We have uploaded a revised version of our paper where these changes are highlighted in red for your convenience.
>
> Beyond these textual revisions, we would like to elaborate on our understandings, which we believe aligns with your observations. **We agree that the DM term has its engine side, but its effectiveness is sufficient only for simple tasks. In complex T2I tasks, the CA term becomes the dominant driver of performance.** Due to the character limit for a single post, we provide our detailed responses in separate posts.
>
> We sincerely appreciate your insightful comments and suggestions. Please do not hesitate to let us know if you have any other questions, or if there are any supplementary experiments you would like us to perform.

---

> ### Author Response · Authors · 2025-11-13
> **Regarding "the conclusion is a little bit conclusive".**
>
> ## We Agree That the DM Term Possesses Certain Engine-Like Capabilities for Few-Step Distillation
>
> We fully believe that the DM term, to some extent, has the ability to independently function as an distillation engine for distillation. The belief is based on the following reasons:
>
> **1. Theoretical Soundness:** From a theoretical standpoint, the distribution matching framework for Inverse Kullback-Leibler (IKL) divergence, as established by works like Diff-Instruct, is solid.
>
> **2. Mechanistic Interpretation:** From a micro-level mechanistic viewpoint, we can revisit the process illustrated in Figure 4(b). While we initially used it to show how DM acts as a regularizer for CA-induced artifacts, the same logic explains its engine-like behavior. The DM loss, in essence, pushes the generator's output towards a direction defined by the difference between the real model's prediction and the fake model's prediction. Since the fake model learns the distribution of the generator's current outputs, this update can be seen as a process that:
> - **Adds information:** It introduces features present in the target distribution (as defined by the teacher model) but missing in the current few-step generated images.
> - **Removes artifacts:** It eliminates features present in the few-step images but absent in the target distribution.
>
> Therefore, it is both intuitive and predictable that this process possesses the inherent capability to enhance the quality of few-step generation.
>
> **3. Empirical Evidence:** Our own experiments support this. As shown in Fig. 2, although there is a significant performance gap between using only DM and only CA, the DM alone setting still makes a difference: it elevates the initially chaotic generation results to a state with semantics. This is particularly evident in the 4-step SDXL results in Fig. 4(b), where the images exhibit plausible color structures and sharp edges. Furthermore, the image quality scores on the benchmark show a clear increase during the early stages.
>
> **In fact, our original submission did not entirely dismiss the capabilities of the DM term, as we noted:**
>
> > _In contrast, **even though it is improper to conclude that the DM term is totally incapable of doing the multi-step to few-step conversion (since in the 4-step experiment it indeed makes relatively reasonable images)**, a significant performance gap exists towards the CA setting..._
>
> **In summary, we agree that for simpler tasks like unconditional CIFAR generation, the success of methods like Diff-Instruct is indeed driven by the DM term itself, independent of any CFG Augmentation.**
>
> ---
>
> ## Despite this capability, relying solely on DM is insufficient to achieve satisfactory results in complex tasks like text-to-image generation. The success of existing methods is primarily driven by the CA term, while DM provides the necessary regularization to stabilize the otherwise unsustainable CA training.
>
> This is the core thesis of our paper. We have elaborated in section 3 that:
>
> - **CA is Necessary for High Performance (Sec 3.1):** Our experiments show that using *only* the DM term yields poor results in T2I tasks. In contrast, using *only* the CA term achieves performance close to the full DMD method. Meanwhile, training with CA alone is unstable and quickly collapses, whereas the combination of CA+DM enables stable training at a high-performance level.
> - **DM is Replaceable (Sec 3.2):** We demonstrate that the DM term can be completely removed and replaced with alternative regularizers—such as a non-parametric mean-variance constraint or a GAN loss—which also successfully stabilize the CA-driven training and maintain high-quality generation.
>
> These findings lead us to a key insight: for complex T2I generation, the **CA term appears to be indispensable** for achieving high performance, whereas the **DM term, while beneficial, to some extent can be replaced or dropped**. This strongly suggests that CA acts as the _primary_ engine in these scenarios, while DM's main role shifts to that of an essential regularizer that enables stable and effective training. While the inherent engine-like properties of DM likely also contribute to the final performance, our experiments indicate that CA is the dominant and non-negotiable component.

---

> ### Author Response · Authors · 2025-11-13
> **Regarding "student model score estimation quality, initialisation, teacher model's score quality, etc..."**
>
> In our work, we strictly followed established practices to control for these variables:
>
> - We initialize the generator, real model, and fake model using the weights of the pre-trained multi-step diffusion model.
> - During training, the real model (teacher) is frozen, while the generator and fake model are optimized.
>
> A particularly noteworthy point that connects to our work is that, if we were to use different learning rates for the generator and the fake model, and set the learning rate of the fake model to zero, the standard DMD loss in fact degenerates into a CA-only setting (as the two terms in the DM subtraction become identical). This helps explain a common empirical observation in DMD training: **the generator's quality improves significantly within the very first few iterations**, long before the fake model has had time to accurately learn the distribution of the few-step generator. Our work now clarifies that this rapid initial improvement is driven by the powerful effect of the CA term.

---

> ### Author Response · Authors · 2025-11-27
> **Follow-up on Rebuttal for Paper Decoupled-DMD**
>
> Dear Reviewer rsWL,
>
> Thank you once again for the valuable time and effort you have dedicated to reviewing our work.
>
> We are truly grateful for your constructive suggestions. We wanted to gently remind you that we have updated our manuscript and provided a point-by-point response addressing the concerns raised in your review. As the deadline for the rebuttal discussion period is approaching, we would greatly appreciate your feedback on our revisions.
>
> We are eager to ensure that our clarifications have resolved your concerns regarding the accuracy of our conclusion. If there are any remaining questions or if further clarification is needed, we are more than willing to engage in further discussion.
>
> Sincerely,
>
> Decoupled-DMD Authors

---

### Official Review · Reviewer_uRhT · 2025-10-30

**Soundness:** 3
**Presentation:** 3
**Contribution:** 3
**Rating:** 8
**Confidence:** 4

**Summary:**

This paper challenges the conventional understanding of the underlying mechanisms of Distribution Matching Distillation (DMD) for distilling pre-trained diffusion models into one/few-step student models. While it might be tempting to think that DMD's success mainly stems from matching the student's output distribution to the teacher's output distribution, the authors decompose the DMD loss into a distribution matching (DM) term and a CFG augmentation (CA) term, arguing that it is the CA term that plays the primary role in the distillation process. Surprisingly, the DM term functions more like a stabilizing regularizer and could be replaced by other regularization terms with different trade-offs. Leveraging this insight, a decoupled noise schedule is proposed for CA and DM to improve the model performance.

**Strengths:**

1. This paper identifies a discrepancy between theory and practice in DMD that CFG is only used in the teacher model but not the student model. This is an interesting observation and a natural motivation for this important research topic.
2. The decomposition of the DMD loss into the DM and CA terms provide novel and valuable insights towards a better and principled understanding of the underlying mechanism of DMD.
3. The arguments and hypotheses in the paper are supported by extensive experiments with ablation studies, demonstrating impressive empirical results.
4. The paper is well-written and easy to understand. It also acknowledges the limitations of the current understanding of the CA term and provides some preliminary discussions.

**Weaknesses:**

Overall, I like the paper very much. My only concern is the paper's claim about the CA term being the engine for DMD, which is a bit strong to me. For example, early DMD papers achieved great distillation performance on unconditional generation for CIFAR images, which is not discussed or explored in this paper.

**Questions:**

Could the authors comment on the issue in the weakness section? One way to address this issue is to reduce the claim to "the CA term is the engine for DMD **in conditional generation**".

---

> ### Author Response · Authors · 2025-11-13
> **Thank you for you comments!**
>
> Dear Reviewer uRhT,
>
> We are deeply grateful for your positive and insightful review. We are particularly encouraged by your positive assessment (rating of 8) and your recognition of our paper's contributions, especially our novel decomposition of the DMD loss and the extensive empirical support.
>
> We especially appreciate your thoughtful critique regarding the strength of our central claim. Your point about the success of early DMD methods on unconditional tasks like CIFAR, where the CA term is absent, is a crucial boundary condition for our hypothesis. You are absolutely right, and we agree that our original claim was too strong and did not adequately specify the scope.
>
> Following your excellent suggestion, we have revised the manuscript to be more precise and better-calibrated. Specifically:
>
> 1. **Refining Our Central Claim:** We have revised our central claim throughout the paper to explicitly state that **"the CA term is the *primary* engine for DMD in complex conditional generation scenarios like text-to-image."** This directly incorporates your feedback and makes our conclusion more rigorous.
>
> 2. **Acknowledging DM's Role:** We have also moderated our language to better reflect the dual role of the DM term. For instance, we now explicitly acknowledge its effectiveness as a standalone engine in simpler, unconditional settings before stating the shifted role towards that of a regularizer in more complex tasks. For instance, phrase "$\mathcal{L}\_{DMD\text{-theory}}$​ usually leads to poor performance in practice" has been revised to "While $\mathcal{L}\_{DMD\text{-theory}}$​ proved to be effective on simple generation tasks like CIFAR, its performance on complex tasks like text-to-image is usually unsatisfactory."
>
> We have uploaded a revised version of our paper with these changes highlighted in red. In essence, our revised position—which we believe aligns with your observation—is that while the DM term can indeed act as a distillation engine (and is the _sole_ engine in unconditional cases), its efficacy is often insufficient for the demands of high-complexity conditional generation. In these complex scenarios, the CA term becomes the dominant driver, and the DM term's primary contribution shifts to that of a crucial regularizer. *We elaborate on our understanding of the success of DM-only distillation for simple tasks like unconditional CIFAR in a separate post.*
>
> Thank you once again for your valuable feedback, which has significantly improved the precision and clarity of our work. Please don't hesitate to let us know if you have further questions.

---

> ### Author Response · Authors · 2025-11-13
> **On the Success of DM-only Distillation for Unconditional CIFAR**
>
> ## We Agree That the DM Term Possesses Certain Engine-Like Capabilities for Few-Step Distillation
>
> We fully believe that the DM term, to some extent, has the ability to independently function as an distillation engine for distillation. The belief is based on the following reasons:
>
> **1. Theoretical Soundness:** From a theoretical standpoint, the distribution matching framework for Inverse Kullback-Leibler (IKL) divergence, as established by works like Diff-Instruct, is solid.
>
> **2. Mechanistic Interpretation:** From a micro-level mechanistic viewpoint, we can revisit the process illustrated in Figure 4(b). While we initially used it to show how DM acts as a regularizer for CA-induced artifacts, the same logic explains its engine-like behavior. The DM loss, in essence, pushes the generator's output towards a direction defined by the difference between the real model's prediction and the fake model's prediction. Since the fake model learns the distribution of the generator's current outputs, this update can be seen as a process that:
> - **Adds information:** It introduces features present in the target distribution (as defined by the teacher model) but missing in the current few-step generated images.
> - **Removes artifacts:** It eliminates features present in the few-step images but absent in the target distribution.
>
> Therefore, it is both intuitive and predictable that this process possesses the inherent capability to enhance the quality of few-step generation.
>
> **3. Empirical Evidence:** Our own experiments support this. As shown in Fig. 2, although there is a significant performance gap between using only DM and only CA, the DM alone setting still makes a difference: it elevates the initially chaotic generation results to a state with semantics. This is particularly evident in the 4-step SDXL results in Fig. 4(b), where the images exhibit plausible color structures and sharp edges. Furthermore, the image quality scores on the benchmark show a clear increase during the early stages.
>
> **In fact, our original submission did not entirely dismiss the capabilities of the DM term, as we noted:**
>
> > _In contrast, **even though it is improper to conclude that the DM term is totally incapable of doing the multi-step to few-step conversion (since in the 4-step experiment it indeed makes relatively reasonable images)**, a significant performance gap exists towards the CA setting..._
>
> **In summary, we agree that for simpler tasks like unconditional CIFAR generation, the success of methods like Diff-Instruct is indeed driven by the DM term itself, independent of any CFG Augmentation.**
>
> ---
>
> ## Despite this capability, relying solely on DM is insufficient to achieve satisfactory results in complex tasks like text-to-image generation. The success of existing methods is primarily driven by the CA term, while DM provides the necessary regularization to stabilize the otherwise unsustainable CA training.
>
> This is the core thesis of our paper. We have elaborated in section 3 that:
>
> - **CA is Necessary for High Performance (Sec 3.1):** Our experiments show that using *only* the DM term yields poor results in T2I tasks. In contrast, using *only* the CA term achieves performance close to the full DMD method. Meanwhile, training with CA alone is unstable and quickly collapses, whereas the combination of CA+DM enables stable training at a high-performance level.
> - **DM is Replaceable (Sec 3.2):** We demonstrate that the DM term can be completely removed and replaced with alternative regularizers—such as a non-parametric mean-variance constraint or a GAN loss—which also successfully stabilize the CA-driven training and maintain high-quality generation.
>
> These findings lead us to a key insight: for complex T2I generation, the **CA term appears to be indispensable** for achieving high performance, whereas the **DM term, while beneficial, to some extent can be replaced or dropped**. This strongly suggests that CA acts as the _primary_ engine in these scenarios, while DM's main role shifts to that of an essential regularizer that enables stable and effective training. While the inherent engine-like properties of DM likely also contribute to the final performance, our experiments indicate that CA is the dominant and non-negotiable component.

---

### Official Review · Reviewer_vHFL · 2025-11-01

**Soundness:** 2
**Presentation:** 3
**Contribution:** 3
**Rating:** 4
**Confidence:** 3

**Summary:**

The paper investigates the respective roles of the two loss components—CFG augmentation and distribution matching—in the DMD framework. Through carefully controlled experiments, the authors conclude that CFG augmentation serves as the primary driver for few-step or one-step conversion, while distribution matching acts mainly as a regularizer. They further argue that, although alternative regularizers could be used, distribution matching remains the best fit. Finally, the paper observes that assigning different $\tau$ values to the two loss terms yields additional performance improvements.

**Strengths:**

* Provides a timely and insightful analysis of the functional roles of DMD’s two loss terms, addressing the open question of why DMD excels in few-step or one-step generation.
* The authors design careful and hypothesis-driven experiments to isolate and test the contribution of each loss term, leading to well-supported conclusions.
* Based on these insights, the paper proposes using distinct $\tau$ values for the two terms, leading to measurable performance gains.

**Weaknesses:**

Most experiments rely primarily on qualitative evaluation (visual inspection of generated images). While visualization is valuable for illustrating effects, heavy reliance on qualitative judgments risks confirmation bias—highlighting supportive examples while overlooking contradictory ones. A more scientifically rigorous approach would involve defining quantitative metrics and validating observations across the entire test set, to ensure statistical robustness and reproducibility.

**Questions:**

See weaknesses.

---

> ### Author Response · Authors · 2025-11-13
> **Thank you for you comments! Rebuttal part1**
>
> Dear Reviewer vHFL,
>
> We sincerely thank you for your time and for providing an insightful review of our work. We are grateful for the acknowledgment of our paper's strengths, particularly in providing a novel analysis of DMD's loss terms and designing hypothesis-driven experiments.
>
> We are happy to address the concern raised in the "Weaknesses" section regarding the perceived over-reliance on qualitative evaluation. We agree that a scientifically rigorous approach requires robust quantitative validation. We believe our work largely adheres to this principle, and the reviewer's concern might stem from the presentation format of some results. To clarify the comprehensive nature of our evaluation, we have summarized all key experiments, their corresponding evaluation methods, and their purposes in the table below:
>
> | Experiment                                | Purpose                                                                                                                | Qualitative Visualization | **Quantitative Evaluation**                                                                                                                                                       |
> | :---------------------------------------- | :--------------------------------------------------------------------------------------------------------------------- | :------------------------ | :-------------------------------------------------------------------------------------------------------------------------------------------------------------------------------- |
> | **1. Ablation Study on CA & DM**          | To demonstrate that CA is the primary driver ("engine") in few-step distillation.                                      | Fig. 2                    | **Fig. 2 (Line Charts)**: Average Image Reward and HPS v2.1 scores on the **COCO-10k dataset**, tracked throughout training.                                                      |
> | **2. CA with Different Regularizers**     | To validate DM's role as a regularizer and further solidify CA's dominant role.                                        | Fig. 8                    | **Fig. 3 (Line Charts)**: Average Image Reward, HPS v2.1, Mean, and Std. on the **COCO-10k dataset**, tracked throughout training.                                                |
> | **3. Mechanistic Analysis of CA & DM**    | To gain an intuitive, microscopic understanding of how CA and DM function, motivating our proposed decoupled schedule. | Fig. 4                    | *None*. This experiment is exploratory and hypothesis-generating by nature.                                                                                                       |
> | **4. Validation of Re-noising Schedules** | To ablatively verify the superiority of our proposed Decoupled-Hybrid re-noising schedule.                             | Fig. 5                    | **Table 1**: Quantitative results on DPG Bench, HPS v2.1, HPS v3. <br> **Sec. C (Tables 3, 4, 5)**: A large-scale **user study** with **15 human annotators** on **500 prompts**. |
> | **5. Comparison with SOTA Methods**       | To demonstrate the state-of-the-art performance of our final proposed method.                                          | Fig. 6, Fig. 7            | **Table 2, Tables 6-17**: Extensive comparisons on **over 10 benchmarks** and dozens of metrics (e.g., FID, CLIP-S, DPG Bench, GenEval, TIIF-Bench, etc.).                        |

---

> ### Author Response · Authors · 2025-11-13
> **Thank you for you comments! Rebuttal part2**
>
> As the table illustrates, **nearly all of our experiments and core claims are substantiated by rigorous quantitative results**.
>
> 1.  The sole exception is **Experiment 3**, which is inherently qualitative. Its purpose is not to definitively prove a hypothesis, but to provide an intuitive, mechanistic insight that *motivates* our proposed decoupled re-noising schedule. The validity of the methodology derived from this insight is then extensively and quantitatively validated in Experiments 4 and 5.
>
> 2.  We speculate the impression "Most experiments rely primarily on qualitative evaluation" might originate from our use of **line charts** in Experiments 1 and 2 (Fig. 2 and Fig. 3). We wish to clarify that these charts are **fully quantitative**. They depict the **dynamic trend of average metric scores** (Image Reward, HPS v2.1, etc.) calculated **across the entire validation dataset**, not the results of a single, cherry-picked image. We chose this format to illustrate how different configurations perform and **evolve** over the course of training, which a single endpoint number cannot capture. We believe this is a robust form of quantitative analysis.
>
> 3.  Furthermore, for our main claim regarding the superiority of the proposed schedule (**Experiment 4**), we went beyond standard automated metrics and conducted a **comprehensive user study** (Sec. C). As detailed in Tables 3, 4, and 5, we collected preference data from 15 professional annotators over 500 prompts. This provides strong, statistically significant evidence from the perspective of human perception, directly addressing the risk of confirmation bias.
>
> In summary, we respectfully believe that our conclusions are built upon a solid and reproducible foundation of large-scale quantitative validation, spanning automated metrics, dynamic training trends, and direct human evaluation.
>
> Furthermore, we appreciate your valuable feedback on data presentation. We agree that tabular results can present the performance metrics at key training steps more clearly and directly. Therefore, in our revision, we will add a summary table for the ablation study results among CA/DM/CA+regularizers to enhance readability. For your convenience, we present the results for this new table below for your reference:
>
> |     Step     |    0    |   200   |   400  |   600  |   800  |  1000  |   2000  |   3000  |   4000  |   5000  |
> |:------------:|:-------:|:-------:|:------:|:------:|:------:|:------:|:-------:|:-------:|:-------:|:-------:|
> | Image Reward |         |         |        |        |        |        |         |         |         |         |
> |      DM      | -223.12 | -199.91 | -67.35 | -38.35 | -62.71 | -45.18 |  -68.62 | -155.62 | -171.71 | -142.38 |
> |      CA      | -223.12 |  -37.70 |  69.94 |  92.97 |  88.32 |  95.10 | -227.64 | -227.73 | -227.87 |    -    |
> |     CA+KL    | -223.12 |  -36.64 |  67.83 |  65.90 |  79.36 |  91.69 |  83.28  |  73.85  |  78.57  |  76.88  |
> |     CA+DM    | -223.12 |  -42.58 |  70.99 |  82.10 |  81.41 |  80.13 |  97.07  |  86.03  |  82.28  |  86.53  |
> |    CA+GAN    | -223.12 |  -21.06 |  83.67 |  83.89 |  97.60 |  97.39 |  100.15 |  95.54  |  67.85  | -227.75 |
> |    HPSv2.1   |         |         |        |        |        |        |         |         |         |         |
> |      DM      |   9.12  |  10.03  |  18.82 |  21.80 |  20.94 |  22.62 |  21.83  |  17.08  |  16.08  |  16.88  |
> |      CA      |   9.12  |  21.44  |  31.03 |  31.27 |  30.69 |  30.88 |   8.73  |   9.35  |   9.47  |    -    |
> |     CA+KL    |   9.12  |  20.89  |  31.01 |  31.45 |  31.57 |  32.01 |  31.05  |  30.49  |  29.98  |  30.15  |
> |     CA+DM    |   9.12  |  20.76  |  30.64 |  30.83 |  31.69 |  31.52 |  32.18  |  32.08  |  31.40  |  31.54  |
> |    CA+GAN    |   9.12  |  22.09  |  32.03 |  31.64 |  32.06 |  32.52 |  32.40  |  31.67  |  29.59  |   7.72  |
>
> We hope this clarification fully addresses your concerns. We thank you again for your valuable feedback, which has helped us improve the clarity of our paper's contributions. We welcome any further concerns you may have and are more than happy to address them.

---

> > ### Comment · Reviewer_vHFL · 2025-11-25
> >
> > Thank you for the detailed discussion, the additional numerical results, and the careful breakdown of qualitative vs. quantitative aspects. The explanation of the line charts in Figures 2 and 3 is very helpful and addresses my main concerns. I will raise my score to 6.

---

> > > ### Author Response · Authors · 2025-11-26
> > > **Thank you for the postive feedback!**
> > >
> > > Dear Reviewer vHFL,
> > >
> > > Thank you very much for your response and for raising the score. We are pleased to know that the detailed discussion and the clarifications on Figures 2 and 3 resolved your concerns. We are grateful for your helpful feedback throughout the review process!
> > >
> > > Decoupled-DMD Authors

---

### Author Response · Authors · 2025-11-29
**AC Letter: Summary of Rebuttal & Discussion for Paper #3330 (Decoupled DMD)**

**Dear Area Chair,**

Due to the recent system revert and the assignment of new ACs, we are writing to provide a concise summary of the consensus reached during the discussion period, as well as the specific remaining questions from the reviewers who have not yet replied.

**1. Score Update (4/8/4 → 6/8/4)**
Our initial scores were 4, 8, and 4. During the discussion, **Reviewer vHFL raised their score from 4 to 6** after we resolved a factual misunderstanding. The other two reviewers did _not_ reply before the system revert.

**2. Resolution with Reviewer vHFL (Score raised 4 → 6)**
The primary concern of vHFL stemmed from a misinterpretation of Figures 2 and 3. The reviewer initially believed these line charts represented statistics for a single sample, leading to concerns about quantitative rigor.

- **Resolution:** We clarified that these charts represent averages over the **entire validation dataset**. We provided the corresponding tabular data to support this.
- **Outcome:** The reviewer acknowledged the misunderstanding was resolved and raised their score.

**3. Reviewers uRhT (Rating 8) and rsWL (Rating 4)**
These two reviewers did not response during the discussion period. Despite the difference in scores, these two reviewers were highly aligned. Remarkably, **both reviewers have only one remaining concern, and it is the exact same concern.**

- **Strengths:** uRhT (8 score) stated, "Overall, I like the paper very much." rsWL (4 score) noted they "really like this topic," found it "very valuable," and agreed the "experiments are sound." Crucially, rsWL noted: "Happy to increase the score if the concern could be solved."
- **The Shared Concern:** Both reviewers pointed out that our conclusion—_"CFG Augmentation (CA) is the engine, Distribution Matching (DM) is the regularizer"_—felt "a bit strong" (uRhT) or "a little bit conclusive" (rsWL). They noted that in simpler domains like low-resolution unconditional CIFAR, DM alone is sufficient to drive generation.

**4. Our Response & Revision: Clarifying the scope to T2I**
We fully agree with the reviewers that we must explicitly scope our conclusions to complex, realistic tasks like **Text-to-Image (T2I)** generation, as the evidence in simple domains (CIFAR) is indeed different. We have made two key responses:

- **The Revision:** We acknowledged that while **DM can serve as an engine for simple tasks (CIFAR)**, it lacks the power for T2I. We have explicitly scoped our claims in the revision to focus on **Text-to-Image generation**, ensuring our conclusions no longer conflict with observations from simpler domains.
- **The Explanation:** Beyond clarifying the scope, we directly addressed how DM functions in simple tasks. We argue that DM indeed possesses a **limited degree** of capability to motivate few-step generation—enough for simple benchmarks like CIFAR. However, this capability is insufficient for the complexity of T2I. Our contribution is identifying that **CFG Augmentation (CA)**—previously seen as a mere "trick"—is actually a robust mechanism capable of **independently** driving distillation in complex scenarios. In summary, while we do not deny the effectiveness of DM on CIFAR-level tasks, we maintain that **in complex text-to-image tasks, its capacity is insufficient to drive the success seen in current DMD practices, whereas CA acts as the primary driver.**

Finally, we believe that both reviewers agree our arguments are sound within the appropriate scope (Text-to-Image), and their concerns were primarily regarding the lack of limiting qualifiers in our expression. We are confident our revision has resolved this.

For your convenience, we have written a separate post that provides a more detailed technical context regarding the background of this work and the specific reviewer questions.

Thank you for your time and for managing this challenging situation.

Best regards,

Authors of Paper #3330

---

> ### Author Response · Authors · 2025-11-29
> **Detailed Context: From CIFAR to T2I**
>
> To ensure clarity regarding the "Engine vs. Regularizer" discussion, we provide this detailed summary of the research context and our specific response to the concern regarding CIFAR vs. Text-to-Image (T2I) dynamics.
>
> **1. Historical Context: The "Trick" that became the Engine**
> Initial works like Diff-Instruct proposed using Distribution Matching (DM) algorithms for few-step diffusion distillation. These prototypes were highly successful in simple scenarios like CIFAR image generation.
>
> However, when DMD attempted to apply this algorithm to realistic **Text-to-Image (T2I)** generation, **it was found to be ineffective**. To solve this, the authors added Classifier-Free Guidance (CFG) to one component (the real model) of the original DM objective, which resulted in satisfactory performance. Subsequent works have widely adopted this operation, treating it as an important "trick" without deeply investigating _why_ it was necessary.
>
> **2. Our Contribution: Decoupling the Mechanism**
> Our work reveals that this so-called "trick" is actually the introduction of a distinct, independent mechanism we term **CFG Augmentation (CA)**, sitting on top of the strict DM mechanism.
>
> Contrary to the conventional understanding, our experiments show a stark contrast in reality:
>
> - **The Reality of T2I:** In complex T2I tasks, CA _alone_ can **independently** achieve high-quality few-step distillation. Conversely, DM _alone_ produces **very limited results**.
> - **The Conclusion:** This indicates that in the successful practice of DMD for T2I, CA is the actual "Engine" driving the few-step capability (Meanwhile, we show that DM acts as a "Regularizer" to prevent the training collapse that occurs in late-stage pure CA training).
>
> **3. Addressing the Reviewer Concern (CIFAR vs. T2I)**
> Because our work focuses on realistic T2I tasks, our initial version stated broadly that "CA is the engine, DM is the regularizer." Reviewers correctly pointed out that this conclusion was too broad, as early research showed DM works fine without CA on CIFAR.
>
> We fully agree with this observation and have clarified our position in the revision and rebuttal:
>
> - **Scope Clarification:** We have explicitly limited our discussion and claims to **complex Text-to-Image tasks**.
> - **Reconciling with CIFAR:** We acknowledge that DM possesses **some** inherent capacity to act as an engine. On simple tasks (CIFAR), this limited capacity is sufficient to produce good results. However, on realistic T2I tasks, this capacity is completely insufficient to match the success of current DMD algorithms.
>
> Therefore, our conclusion that "CA is the engine" is specific to the context of modern, high-complexity generation tasks. We believe this distinction resolves the concern while preserving the core contribution of our analysis.
>
> Best regards,
>
> The Authors

---

### Meta-Review · Area_Chair_n64r · 2026-01-08

**Summary:**

This paper studies Distribution Matching Distillation (DMD), proposing that its success in text-to-image (T2I) generation stems from decomposing the objective into two parts: 1. CFG Augmentation (CA) which acts as the primary engine for few-step generation, 2. Distribution Matching (DM) which acts as a regularizer for stability. Based on this, the authors propose a decoupled re-noising schedule that improves performance. The reviewers generally found the decomposition insightful and the topic valuable. The primary concerns focused on the scientific rigor of the evaluation (the interpretation of visual metrics) and the scope of the authors' claims regarding DM, particularly given that DM works as a standalone engine in simpler domains like unconditional CIFAR. As the authors effectively clarified the evaluation metrics and appropriately scoped their claims to complex T2I tasks during the rebuttal, my suggested decision is to Accept

**Reviewer Concerns:**

*** Addressed

Quantitative Rigor (Reviewer vHFL): Reviewer vHFL initially criticized the paper for relying too heavily on qualitative visualization and misinterpreted the line charts in Figures 2 and 3 as representing single samples rather than dataset averages. The authors clarified that these charts represented averages over the validation set and provided tabular data. They also pointed to their user study. The reviewer explicitly acknowledged that this resolved their main concern.

Scope of the Engine vs. Regularizer Claim (Reviewers uRhT and rsWL): Both reviewers argued that the claim "CA is the engine, DM is the regularizer" was too strong because previous works (like Diff-Instruct) showed DM works independently on simpler tasks like unconditional CIFAR. The authors conceded this point, agreeing that DM has engine-like capabilities in simple settings but arguing it is insufficient for complex T2I. They revised the manuscript to explicitly scope their claims to complex conditional generation scenarios like text-to-image. This revision directly addresses the scientific validity of the claim.

*** Outstanding

While Reviewers uRhT and rsWL did not reply to the final rebuttal post to confirm the text changes were sufficient, the authors' revision aligned with the reviewers' specific requests.

**Reviewer Scores:**

Reviewer vHFL: Score changed from 4 to 6. The reviewer explicitly raised their score after the misunderstanding regarding the quantitative plots was resolved.

Reviewer uRhT: Score would likely remain 8. This reviewer was already very positive and their only weakness was the scope of the claim, which the authors addressed.

Reviewer rsWL: Current score is 4 and the reviewer stated they were "Happy to increase the score if the concern could be solved". Since the concern was identical to uRhT's (the CIFAR/unconditional counter-example) and the authors implemented the specific scoping requested, I believe the reviewer would have aligned their score with uRhT had they participated in the final discussion.

---

### Decision · Program_Chairs · 2026-01-26

Accept (Poster)